# Striking circadian neuron diversity and cycling of *Drosophila* alternative splicing

Qingqing Wang[1,2,3†], Katharine C Abruzzi[4,5†]*, Michael Rosbash[4,5], Donald C Rio[1,2,3]*

[1]Department of Molecular and Cell Biology, University of California, Berkeley, Berkeley, United States; [2]Center for RNA Systems Biology (CRSB), University of California, Berkeley, Berkeley, United States; [3]California Institute for Quantitative Biosciences (QB3), University of California, Berkeley, Berkeley, United States; [4]Department of Biology, Howard Hughes Medical Institute, Brandeis University, Waltham, United States; [5]National Center for Behavior Genomics, Brandeis University, Waltham, United States

**Abstract** Although alternative pre-mRNA splicing (AS) significantly diversifies the neuronal proteome, the extent of AS is still unknown due in part to the large number of diverse cell types in the brain. To address this complexity issue, we used an annotation-free computational method to analyze and compare the AS profiles between small specific groups of *Drosophila* circadian neurons. The method, the Junction Usage Model (JUM), allows the comprehensive profiling of both known and novel AS events from specific RNA-seq libraries. The results show that many diverse and novel pre-mRNA isoforms are preferentially expressed in one class of clock neuron and also absent from the more standard *Drosophila* head RNA preparation. These AS events are enriched in potassium channels important for neuronal firing, and there are also cycling isoforms with no detectable underlying transcriptional oscillations. The results suggest massive AS regulation in the brain that is also likely important for circadian regulation.
DOI: https://doi.org/10.7554/eLife.35618.001

*For correspondence:
katea@brandeis.edu (KCA);
don_rio@berkeley.edu (DCR)

†These authors contributed equally to this work

**Competing interests:** The authors declare that no competing interests exist.

## Introduction

Organisms ranging from cyanobacteria to mammals contain circadian clocks that synchronize physiology and behavior to time-of-day environmental cues, such as light and temperature. The fruit fly, *Drosophila,* is no exception and exhibits robust circadian clocks that control a diverse range of behaviors such as activity, sleep, mating and feeding. The core molecular machinery of the circadian clock is a set of transcription factors that act together to drive cycling or time-of-day dependent transcription of at least 20% of genes in the genome (*Abruzzi et al., 2011*). The heterodimeric transcription factor CLOCK/CYCLE (CLK/CYC) binds to the promoters of many direct target genes and activates their transcription late in the day. Two of these direct target genes, *Timeless* (*Tim*) and *Period* (*Per*), encode transcriptional repressors that re-enter the nucleus in the late night and act as part of a negative feedback loop to turn off CLK/CYC transcription (*Edery et al., 1994*; *Naidoo et al., 1999*; *Sehgal et al., 1994*). In the early morning, the photoreceptor, Cryptochrome (CRY) and the kinase, Doubletime (Dbt) contribute to the degradation of Tim/Per (*Kim et al., 2007*; *Kloss et al., 1998*; *Emery et al., 1998*; *Stanewsky et al., 1998*; *Ceriani et al., 1999*; *Hunter-Ensor et al., 1996*; *Syed et al., 2011*). Without these repressors, CLK/CYC-mediated transcription begins again restarting the daily cycle of transcription.

In the *Drosophila* brain, this molecular clock resides in a neural network of ~150 neurons. These neurons are divided into seven subgroups including three groups of dorsal neurons (DNs; DN1, DN2, and DN3), three groups of lateral neurons (ventral and dorsal lateral neurons; large and small

**eLife digest** The life of nearly all creatures on Earth follows the rhythm of day and night. For example, in fruit flies, darkness and light dictate when the insects feed, rest, move or mate. This is possible thanks to the circadian clock, an internal program which is synchronized with the environment to tell cells in the body when to perform certain roles.

In fruit flies, the structure that keeps the body clock ticking is formed of about 150 'circadian neurons', which are divided into several subgroups. In these cells, a complex genetic programis at work, with networks of genes being 'switched on' in a cyclical way. To understand how this program works, scientists need to know which genes are turned on and when, as well as which proteins are created based on the information contained in these genes.

This can be difficult because one gene does not necessarily code for only one protein. Indeed, when a gene is turned on, it gets copied into a pre-messenger RNA (pre-mRNA), which is formed of several modules. The pre-mRNA can then go through a process called alternative splicing that shuffles or removes the different modules. This means that one gene can give rise to different pre-mRNA molecules that will each serve as a template to build a distinct protein. Until now, there have been few studies that examine the different types of pre-mRNAs found in circadian neurons, and how these change with the time of day.

Here, Wang, Abruzzi et al. extract three subgroups of circadian neurons, and one subgroup of non-circadian neurons, from the brain of fruit flies. The pre-mRNAs are isolated, and then a new computational method, called JUM, identifies, counts and categorizes the pre-mRNA molecules in the different groups of neurons.

This analysis reveals hundreds of previously unknown pre-mRNA molecules, many of which differed extensively between the types of brain cells. When comparing circadian and non-circadian neurons, Wang, Abruzzi et al. show that the circadian cells had more pre-mRNAs that code for proteins that help the cell communicate with other neurons. Finally, many genes in the circadian neurons use alternative splicing to turn on different types of pre-mRNA molecules at different times of the day in a cyclical way; this suggests that these pre-mRNAs might be participating in the genetic circadian program.

Many human disorders, such as certain forms of insomnia, emerge when the circadian clock is thrown off balance. The results reported by Wang, Abruzzi et al. show that alternative splicing may be an overlooked mechanism that shapes this complex program.

DOI: https://doi.org/10.7554/eLife.35618.002

LNvs and LNds, respectively) and the lateral posterior neurons (LPNs) with discrete behavior functions; reviewed in (*Peschel and Helfrich-Förster, 2011*). These neurons make up a small portion of the Drosophila brain (~0.1%) and head (~0.05%). As a result, it is difficult to profile these specialized neurons as part of studies that focus on Drosophila brain and head tissues (*Hughes et al., 2012*; *Rodriguez et al., 2013*).

To learn more about the function of these neurons within the circadian neuronal circuit, three subgroups (DN1s, LNds, and LNvs) of these neurons as well as one non-circadian outgroup (Dopaminergic or tyrosine-hydroxylase (TH) expressing neurons) were labeled with GFP using neuron-specific drivers and manually isolated (*Abruzzi et al., 2015*; *Abruzzi et al., 2017*). Illumina RNA-seq datasets from these neurons were analyzed to examine the global mRNA landscape at the transcription level. This analysis revealed many cycling transcripts whose abundance changes with time of day that were not identified in previous studies of brain or head tissues (*McDonald and Rosbash, 2001*; *Claridge-Chang et al., 2001*; *Wijnen et al., 2006*; *Rodriguez et al., 2013*). In addition, many neuron-specific transcripts including novel circadian neuropeptides were revealed.

Besides transcriptional level gene regulation, post-transcriptional level regulation, such as alternative pre-mRNA splicing, is also known to be essential for the normal functioning of the nervous system. Alternative pre-mRNA splicing (AS) is a major gene regulatory mechanism that enables a single gene locus to produce populations of often functionally distinct mRNAs in a tissue- or cell-type-specific manner, greatly diversifying the metazoan transcriptome. The nervous system is well recognized to exhibit extraordinarily complex and diverse AS patterns in a variety of metazoan organisms

(*Li et al., 2007*; *Wang et al., 2008*; *Irimia et al., 2014*; *Li et al., 2015*). Importantly, these diversified AS patterns have been shown to play important roles in modulating numerous neuronal states and activities (*Vuong et al., 2016*; *Norris and Calarco, 2012*). For example, many neuronal receptors such as the NMDA receptors undergo AS that results in functionally distinct proteins that are regulated in different ways to directly affect neuronal membrane depolarization and action-potential firing (*Chandrasekar, 2013*).

Recent findings reported several cases where AS is involved in the modulation of the circadian clock in plants, mice and *Drosophila* (*Sanchez et al., 2010*; *Petrillo et al., 2011*; *McGlincy et al., 2012*; *Hughes et al., 2012*; *Preußner et al., 2014*). However, a comprehensive understanding of AS profiles and their role in circadian clock regulation remains largely unexplored. As most studies profiling the *Drosophila* neuronal transcriptome originate from head samples, it has been difficult to detect cycling of alternatively spliced transcripts because heterogeneous tissues can often mask cell-type specific transcript cycling (*Kula-Eversole et al., 2010*; *Nagoshi et al., 2010*; *Abruzzi et al., 2017*). In addition, currently most AS analysis software for short-read Illumina RNA-seq data depend on pre-annotated libraries of known spliced transcripts, making it difficult to detect and quantitate novel neuronal AS events that prevail in neuronal tissues.

Here, we have comprehensively analyzed the global AS profiles in the transcriptomes of manually dissected DN1, LNd, and LNv circadian neurons, as well as the non-circadian, TH control neurons described above, using a new computational tool called JUM (the Junction Usage Model) (*Wang and Rio, 2017a*). JUM is able to identify, quantitate and categorize tissue-specific AS patterns from RNA-seq datasets without the need for or use of prior genome or transcriptome annotations. Using JUM, we identify hundreds of previously unannotated pre-mRNA isoforms. They differ extensively among the four neuronal subgroups and were totally missed when RNA-seq data from the heterogeneous fly head sample was analyzed using identical methods. Gene ontology analysis of differentially spliced mRNAs in circadian neurons versus non-circadian neurons revealed that they are enriched for transcripts encoding potassium channels, a hallmark of neuronal excitability. In addition, we discovered many new alternatively spliced variants and cycling AS patterns in transcripts encoding proteins of the molecular clock including the photoreceptor Cryptochrome and the kinase Shaggy. Importantly, we also identified a large set of transcripts that exhibit cycling patterns of alternative splicing throughout the day-night cycle, potentially contributing to the normal functioning of the circadian clock. This study highlights and reinforces the idea that post-transcriptional processes like AS impact cell type specification, diversity and function in the brain.

## Results

### There are many novel AS patterns in neuron subpopulations

To investigate how AS regulation differs with circadian neuron identity and function, we compared the pre-mRNA splicing profiles from three small groups of circadian neurons, the DN1s, LNds and LNvs, as well as from a non-circadian outgroup, the dopaminergic or tyrosine hydroxylase-expressing (TH) neurons. Total RNA and then mRNA from ~100 manually isolated neurons were purified from entrained flies every four hours across two days (12 samples for each group). A mixture of oligo-dT and random-primed cDNA was used to create RNA-seq libraries from each sample, as previously described (*Abruzzi et al., 2015*).

To profile the AS patterns from each neuron group, we first collapsed the time-series RNA-seq data for each neuron group into two pooled datasets, one for each day, and compared the profiles of alternatively spliced junctions using JUM (*Wang and Rio, 2017a*). We chose the JUM software because of its unique ability to identify, categorize and quantitate global splicing patterns without any *a priori* knowledge of, or need for, a genome or transcriptome annotation. Since neurons often exhibit exceptionally diverse AS patterns that are not documented in current transcriptome annotations (*Li et al., 2007*; *Wang et al., 2008*; *Irimia et al., 2014*; *Li et al., 2015*), JUM allows for the discovery of novel and unannotated splicing events and patterns. JUM exclusively utilizes RNA-seq reads that map to splice junctions for AS analysis and defines alternatively spliced junctions (AS junctions) as splice junctions that have alternative 5' - or 3'- splice sites (*Figure 1A*; left panel).

To further evaluate the tissue-specificity and diversity of AS from these neuron groups, we compared their profiles of AS junctions to those from a separately prepared *Drosophila* head sample,

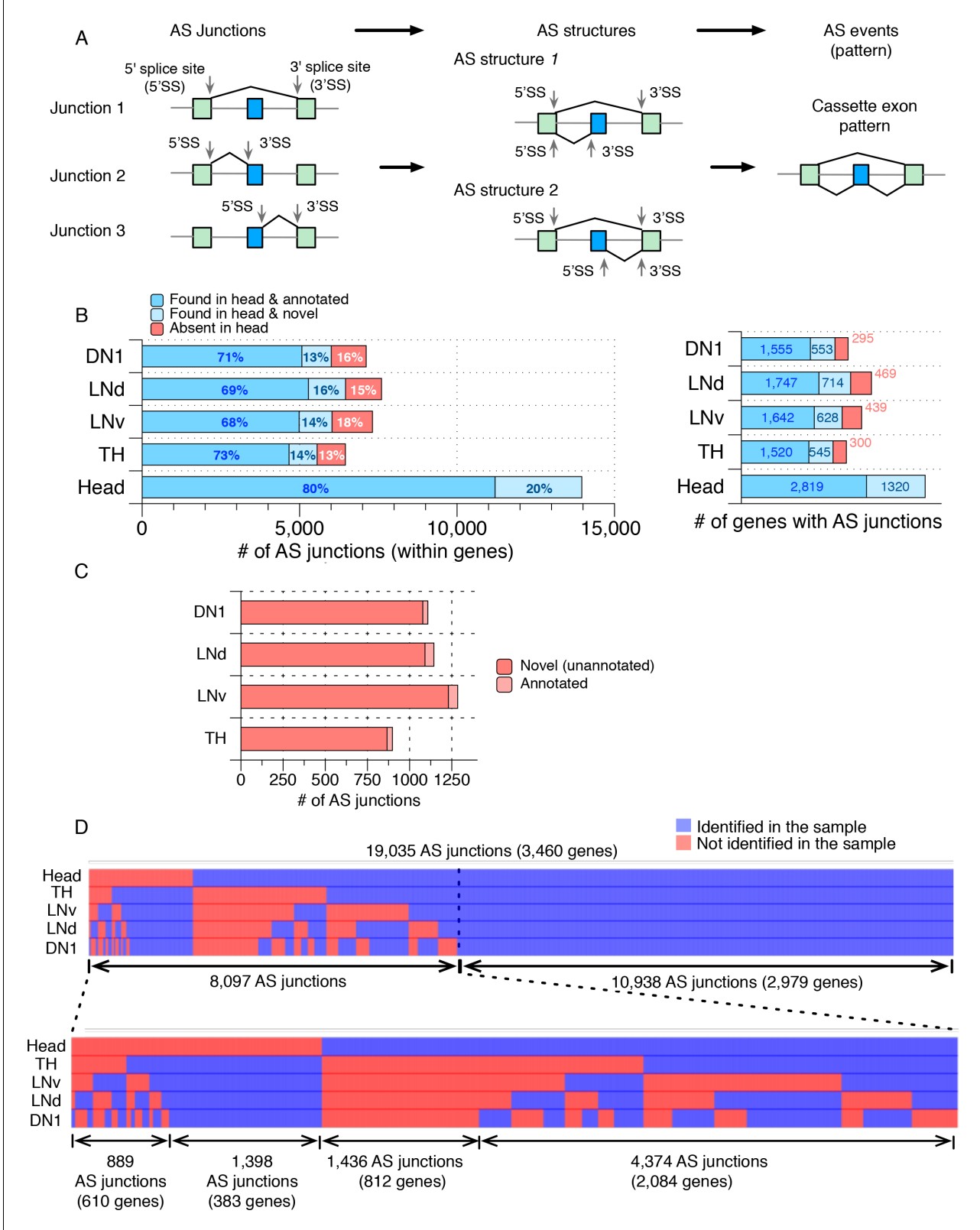

**Figure 1.** Many novel alternatively spliced junctions were identified in isolated *Drosophila* neurons. Alternatively spliced junctions (AS junctions) were identified in RNA-seq libraries generated from isolated *Drosophila* neurons (DN1s, LNds, LNvs, and dopaminergic (TH)) as well as from a separately prepared heterogeneous *Drosophila* whole head sample. (**A**) Pictorial representation of the Junction Usage Model (JUM), using a cassette exon AS pattern as an example. AS junctions are identified in RNA-seq data from reads that have alternative 5'- or 3'- splice sites. These AS junctions are

*Figure 1 continued on next page*

*Figure 1 continued*

grouped into AS structures, defined as a set of splice junctions that share the same 5'- or 3'- splice site. The usage of each sub-AS-junction, that is the relative abundance of that sub-AS-junction compared to the other AS junctions in the same AS structure is calculated and can be compared across different conditions for differential AS analysis. AS structures are then assembled into conventionally recognized AS patterns based on the unique topological feature of the splice graph representing each pattern. (B) The bar graph on the left shows the number of AS junctions identified in each sample, including only the ones that are found within a gene. Darker blue indicates those AS junctions found in the heterogeneous whole head transcriptome and also annotated in the current genome annotation. Lighter blue denotes those junctions identified in the whole head transcriptome and not in the current genome annotation. Red denotes those junctions found in isolated neurons but not in the whole head transcriptome. The bar graph on the right shows the number of genes with the corresponding ALT junctions as specified on the left. (C) The majority of the AS junctions exclusively identified in neuron subtypes are novel and not annotated in the current genome annotation (~95%; shown in red). (D) Heat map of all AS junctions identified from the neuronal subtypes and the whole head sample. Every column is an AS junction and if the junction is identified in a sample, a blue grid is shown; otherwise a red grid is shown.

DOI: https://doi.org/10.7554/eLife.35618.003

The following source data and figure supplements are available for figure 1:

**Source data 1.** AS junctions identified in DN1, LNd, LNv, TH neurons and heads.

DOI: https://doi.org/10.7554/eLife.35618.006

**Figure supplement 1.** Many AS junctions with varying lengths were identified in isolated neuron subtypes.

DOI: https://doi.org/10.7554/eLife.35618.004

**Figure supplement 2.** Transcripts with neuron subgroup specific novel ALT junctions reflect critical neuronal functions.

DOI: https://doi.org/10.7554/eLife.35618.005

the most common source of adult nervous system RNA. We identified tens of thousands of AS junctions in each distinct neuronal sample with extensive length variation (*Figure 1—figure supplement 1*). Importantly, almost all AS junctions contain canonical splice sites (*Supplementary file 1*). Below, we will focus our description on the events that are typical of canonical AS junctions, which are located within single genes.

We identified a total of ~6,000 AS junctions in each of the distinct neuronal subtypes that reside in ~2000 genes, and ~14,000 AS junctions in heads that reside in ~3000 genes (*Figure 1B*; *Figure 1—source data 1*; see Materials and methods). Approximately 20% of these head AS junctions are novel, that is, they were unannotated in the current *Drosophila* genome annotation, and these novel AS junctions are found in 1320 genes (*Figure 1B*; light blue). Importantly, many AS junctions were identified exclusively in each of the neuronal subpopulations compared to fly heads (~2300 in total, in 839 genes; *Figure 1B*; red), and nearly all of them (~95%) were novel, that is, they had not been previously annotated in the current *Drosophila* genome annotation (genome version: FB2017_05; *Figure 1C*). This analysis provides a glimpse of the transcriptome diversity in these neuron groups, which is undetectable in total fly head RNA.

Comparing the AS junctions in the DN1, LNd, LNv, TH neurons and heads identifies not only many AS junctions that are specific to a particular neuronal group but also many with extensive overlap among the neuron subgroups (*Figure 1D*). Indeed, approximately 60% of the novel AS junctions (1398 within 383 genes) are present in all 4 groups of neurons but not present in heads (*Figure 1D*; bottom left). These may be common splice junctions of transcripts that are not ubiquitous and therefore less abundant and not detectable in head transcriptomes. The remaining 40% of the novel AS junctions (889) are specific to one or more of the neuron groups (*Figure 1D*; bottom leftmost) and are from 610 genes. These junctions may therefore specify cell-type-specific protein isoforms resulting in functional differences between the neurons. Another 1,436 AS junctions were identified in fly heads but not in any of the four neuronal groups (*Figure 1C,D*) and are from 812 genes; these junctions may be non-neuronal, for example, from glia, or they may derive from neurons other than the four groups characterized here. Finally, 4,374 AS junctions are identified in fly heads as well as in a subset of the neuronal groups and are from 2084 genes (*Figure 1D*; bottom rightmost).

Importantly, these identified novel AS junctions are predicted to encode protein isoforms that may impact the function of their specific neuron groups. Gene ontology analysis reveals that the novel AS junctions impact crucial functions in each of these neuron subgroups including neurotransmitter secretion in LNvs, acetylcholine-activated cation-selective channel activity (covering five different subunits of the nicotinic acetylcholine receptors) and synaptic target recognition in LNds,

potassium ion transporters and locomotor rhythms in DN1s, and chemical synaptic transmission in THs (*Figure 1—figure supplement 2*; *Supplementary file 2*).

One of the circadian genes that shows a novel AS junction in DN1s is the circadian photoreceptor cryptochrome (cry). cry has a novel splice junction in DN1 and LNd neurons, which is absent in the LNv, TH and head samples (*Figure 2A*; the sashimi plot indicates the number of sequencing reads spanning the splice junctions [*Katz et al., 2015*]). This novel AS junction spans from the 5'- splice site right after the first exon to an alternative 3'-splice site in the first intron of cry and results in a truncated, short transcript and presumptive short protein isoform that lacks the functional domains of cry. Approximately 20% of the DN1 (*Figure 2A*) and approximately 28% of the LNd (*Figure 2A*) cry transcripts are spliced into this short transcript isoform. Furthermore, CG10483 transcripts encoding a putative G-protein-coupled receptor (GPCR) experience a novel exon skipping event (*Figure 2B*; skipped exon marked by '*') only in LNds and DN1 (*Figure 2B*). This event cleanly removes ~65 aa from the receptor transmembrane regions in the middle of the protein.

## Circadian neurons present specific alternative pre-mRNA splicing patterns

To explore the potential regulatory role of alternative pre-mRNA splicing in circadian rhythms, we compared the global AS patterns in the three subtypes of circadian neurons (DN1, LNd and LNv) with the non-circadian TH neurons using JUM. To do this, JUM grouped all identified AS junctions into the basic AS quantitation unit called AS structures, which are a set of AS junctions that share the same 5' splice site or 3' splice site (*Figure 1A*; middle panel). Each AS junction within an AS structure is defined as a sub-AS-junction of the corresponding AS structure. JUM then quantifies the 'usage' of each sub-AS-junction in every AS structure, that is the relative abundance of each sub-AS-junction in the AS structure and identifies AS structures whose usage of sub-AS-junctions are significantly different between the circadian and non-circadian neuron groups. Finally, JUM faithfully assembles the AS structures into the conventionally recognized AS patterns (cassette exon - SE, alternative 5' or 3' splice site – A5SS/A3SS, mutually exclusive exons - MXE, and intron retention - IR) based on the unique topological features of each pattern (*Figure 1A*; right panel) (*Wang and Rio, 2017a*). While our first analysis identified 'all or none' neuronal group-specific splice junction usage, this approach allows for the identification of differential splice junction usage in which AS junctions are present in both neuronal subgroups but used at significantly different frequencies.

Using this method, we found 249, 194 and 70 AS events that are significantly differentially spliced in the DN1, LNv and LNd neurons compared to the TH neurons, respectively, which cover all five conventionally classified AS patterns (*Figure 3A*; Materials and methods; *Figure 3—source data 1–3*). To test whether the differentially spliced AS events identified are specific to a particular group of neurons, we examined the overlap of the differentially spliced AS events in each neuronal subgroup (*Figure 3B*, top right; *Figure 3—figure supplement 1*). Remarkably, the majority of the differentially spliced AS events in DN1, LNv or LNd neurons, compared to TH neurons, are unique to each of the circadian neuronal subtypes (*Figure 3B*). Only approximately 1.6–6.6% (9-36) of these differentially spliced AS events were found to overlap between 2 of the three circadian neuronal subgroups, which is significantly lower than expected (Hypergeometric Test; p value = 2.38e-07 for DN1/TH and LNd/TH; p value = 3.68e-39 for DN1/TH and LNv/TH; p value = 1.7e-04 for LNd/TH and LNv/TH). Only 1.8% (8) of these differentially spliced AS events are found in all three circadian neuron groups. Three of these eight genes are involved in regulating neuronal plasticity, remodeling and synaptic transmission (Pten, Sap47, Rim). This result suggests that each circadian neuronal subgroup possesses a unique pattern of AS isoforms that contribute to the identity of each distinct neuronal group (see Discussion). To further support this conclusion, we also directly compared differential AS patterns within the circadian neuron subgroups; as predicted by the above results, we found many neuronal group-specific AS events (*Figure 3—figure supplement 2*; *Figure 3—source data 4–6*).

To explore mechanisms that might contribute to these circadian neuron group-specific AS profiles, we identified RNA binding proteins (RBPs) that are differentially expressed in each of the circadian neurons compared to the non-circadian TH neurons (*Figure 3B*; *Figure 3—source data 7–9*). Interestingly, each circadian neuron subpopulation expresses its own unique set of differentially expressed RBPs compared to TH neurons, with very limited overlap (*Figure 3B*, lower left panel).

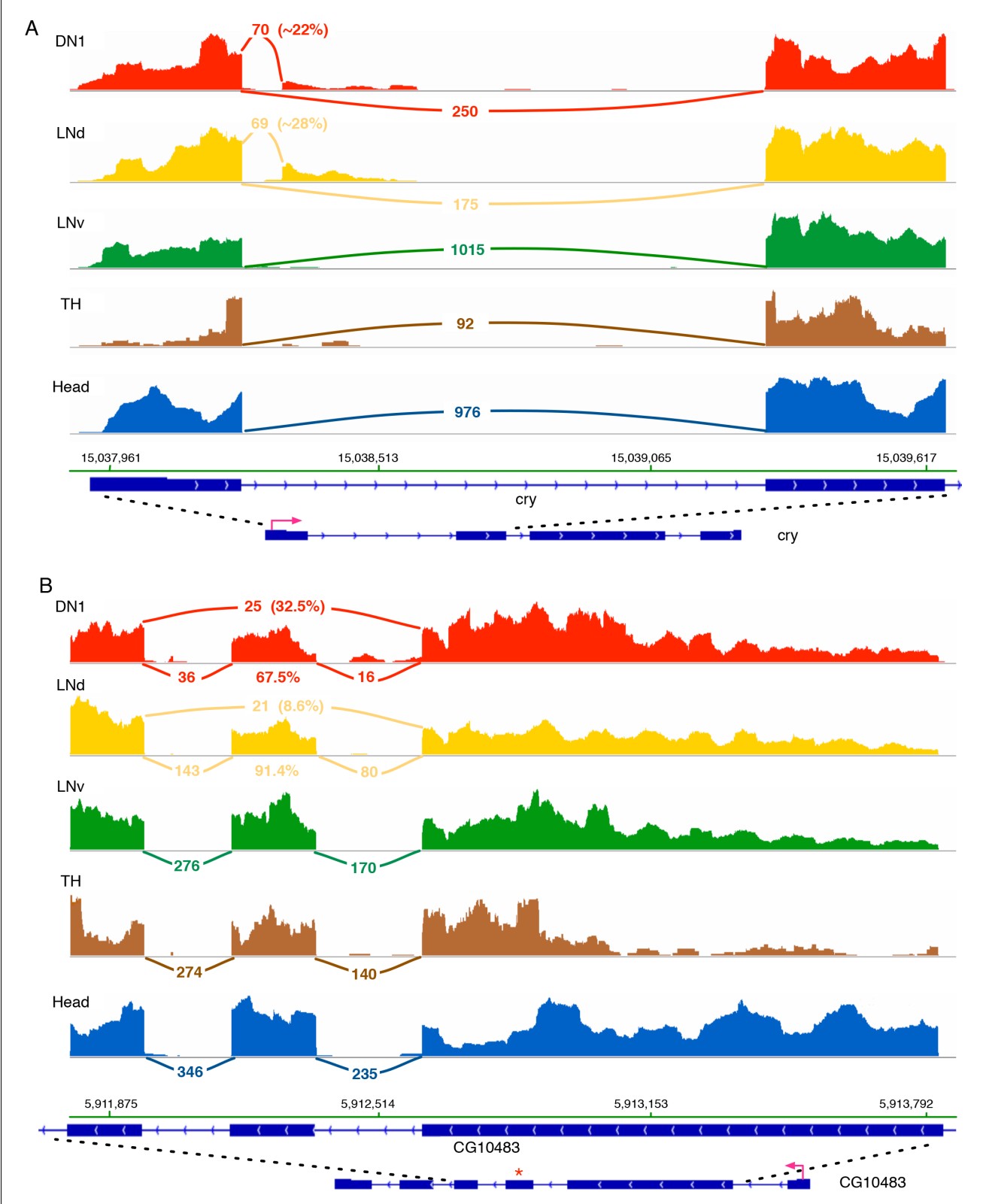

**Figure 2.** *Cry* and *CG10483* transcripts show neuron subtype-specific alternative splicing that is absent in the heterogeneous head sample. Neuron subtype-specific novel alternative splicing of *cry* and *CG10483* are shown. RNA-seq data tracks derived from the neuronal samples are shown, with arcs representing splice junctions and the number of unique-mapped RNA-seq reads mapped to the junction across the arc. A percentage number is shown to indicate the usage of the specific junction or the corresponding splicing isoform. The orientation of the transcript is shown at the bottom: the red

*Figure 2 continued on next page*

*Figure 2 continued*

arrow indicates the direction of the promoter. The dotted lines indicate the region of transcript that is enlarged to highlight the alternatively spliced region. (A) The transcript of the blue-light photoreceptor, cry, has a novel alternative splicing junction in DN1s and LNds that is not observed in the head transcriptome: the first exon is spliced to a novel exonic region within the first intron. (B) The putative G-protein coupled receptor, CG10483, undergoes an exon-skipping event in LNds and DN1s that is not observed in whole heads. The skipped exon three is marked by '*' and is skipped in ~10% of the CG10483 transcripts in LNds and ~30% of the CG10483 transcripts in DN1s.

DOI: https://doi.org/10.7554/eLife.35618.007

Further profiling of the identified targets of a subset of these RBPs using publicly available CLIP and RIP-seq experimental data (*Stoiber et al., 2015*; *Hansen et al., 2015*) suggest that these specific RBPs could account for many AS events. For example, Syb and mub, two RBPs that are differentially expressed in LNv neurons compared to TH neurons, target 23 and 32 LNv-specific differentially spliced AS target RNAs in LNv versus TH neurons, respectively, with five overlaps, which covers ~36% of total LNv neuron specific differentially spliced AS events. Although speculative, this analysis highlights the possibility that circadian neuron-specific RBP expression could drive much of the observed circadian neuron-specific AS patterns compared to TH neurons.

One example of a differentially spliced transcript in LNv neurons compared to TH neurons is the neuronal synaptobrevin (N-syb) transcript. It shows a much higher inclusion frequency of an exon (*Figure 4A*; marked by '*') present in N-syb variant J. 85% of n-Syb transcripts in LNv cells utilize this exon compared to 25–36% in LNd, DN1 and TH neurons (*Figure 4A*). Inclusion of this exon should give rise to a unique protein of 206 amino acids containing an alternative 85 bp C-terminus.

The potassium channel Shab also undergoes alternative 3'-splice site use. In TH and LNd neurons the Shab transcripts preferentially utilize an alternative terminal exon (*Figure 4B*; marked by '*'). It results in a significantly shorter isoform and encodes a protein lacking the sixth transmembrane region of the potassium channel protein (*Figure 4B*), which function as a dominant negative protein. 61–74% of Shab transcripts encode this short isoform in TH and LNd neurons, whereas with the ratio of the short isoform decreases significantly, to 10 and 40%, in LNvs and DN1s (*Figure 4B*).

To further investigate which molecular functions in circadian neurons might be impacted by AS, we performed gene ontology (GO) analysis (*Figure 5A*) on those genes with significantly differentially spliced AS events in each of the circadian neuronal subtypes. Interestingly, potassium ion transport was the top GO term for both the DN1 and LNv neurons (*Figure 5A and B*). The differentially spliced AS events include six different potassium channels, as well as two sodium-potassium-exchanging ATPases (see Shab (above) and *Figure 5C*). These ion channels modulate neuronal excitability and many of them have been implicated in controlling both circadian locomotor activity and sleep (*Jaramillo et al., 2004*; *Cirelli et al., 2005*; *Pimentel et al., 2016*; *Fogle et al., 2015*, see Discussion).

Other notable GO terms in the LNv neuronal population are neuronal projection (*Fas2*, *unc-13*, *Slo*, *Dscam1*, *Stai*), terminal bouton (*cpx*, *Rab3-GEF*, *Syn*, *unc-13* and *n-syb*) and synaptic vesicle exocytosis (*Tomosyn*, *cpx*, *syn*, *unc-13*, *Rim*). These GO terms capture some of the unique features of the LNv neuronal population: they exhibit highly dynamic neuronal projections that undergo time-of-day dependent morphological changes and synaptic vesicle localization (*Fernández et al., 2008*; *Gorostiza et al., 2014*; *Petsakou et al., 2015*). Other notable GO terms for the DN1s include the calcium-calmodulin-dependent family of protein kinases (*CASK*, *CAMKI*, *CAMKII*, and *CG17528*), as well as sodium ion transporters (*Nhe3*, *Nckx30c*, *NaCP60E*, *Atpalpha*, *nrv3*). In contrast, the lower number of differentially alternatively spliced transcripts in the LNds resulted in no statistically significant GO terms.

To further explore the functional importance of the differentially spliced AS events in circadian neurons versus TH neurons, we examined the cross-species conservation of identified differentially spliced cassette exons and the preservation of reading frame from the AS of these cassette exons. We first plotted the PhastCons conservation score (*Siepel et al., 2005*) across 27 species of insects for the sets of alternatively spliced and non-alternatively spliced cassette exons, respectively, in each of the circadian neuron subgroup versus TH comparisons (*Figure 3—figure supplement 3A*). Interestingly, the alternatively spliced cassette exons are somewhat more conserved than the non-alternatively spliced cassette exons, although the result is not statistically significant (Mann-Whitney U Test; p value 0.17). We also found that the inclusion/exclusion of the alternatively spliced cassette exons

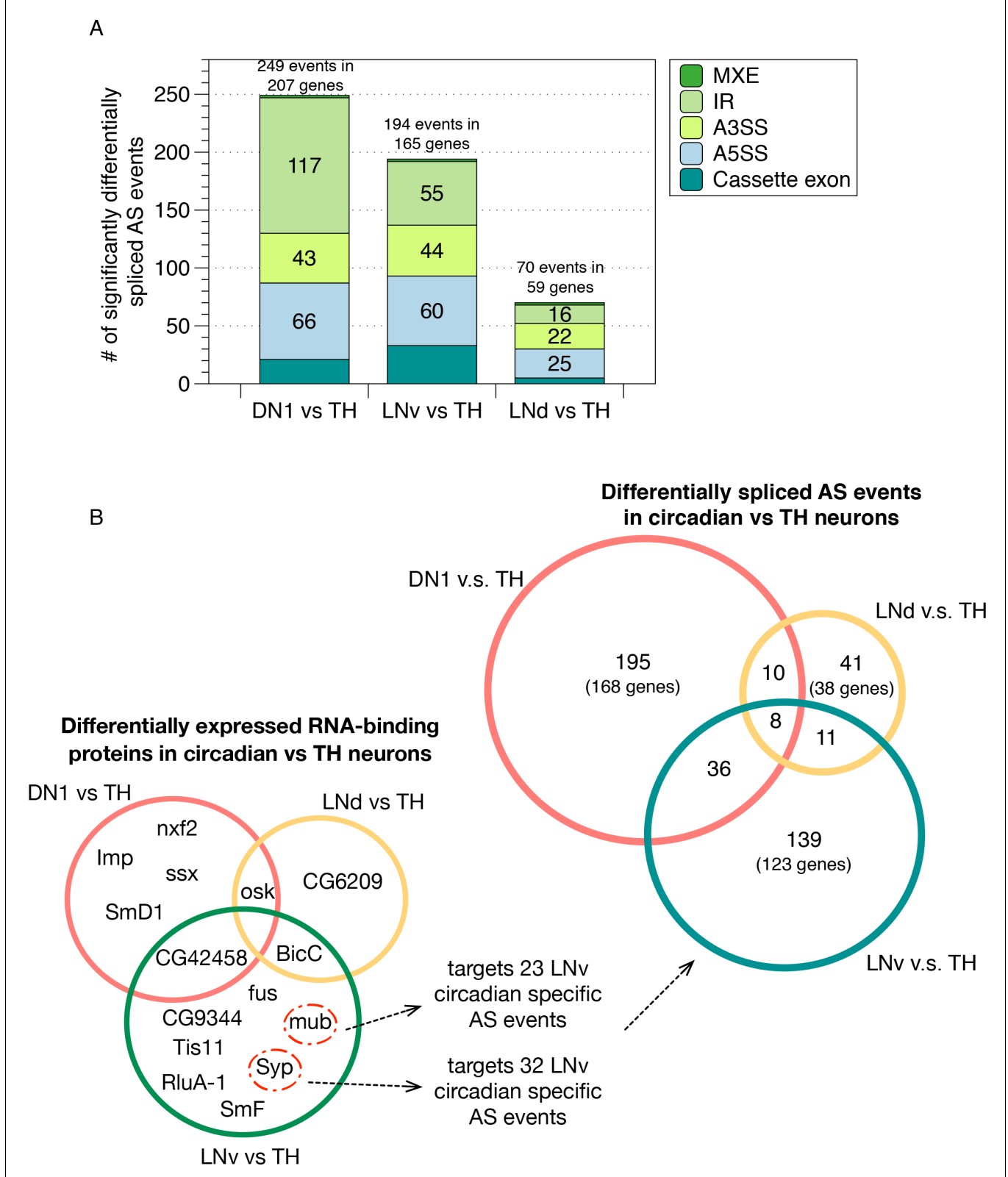

**Figure 3.** Identification of differentially spliced AS events in circadian neurons compared to non-circadian, TH neurons. (A) AS events that were differentially spliced in the circadian neurons (DN1, LNd and LNv) compared to the dopaminergic neurons (TH) were identified using JUM, respectively. The number and type of these differentially alternatively spliced AS events are shown. (Cassette exon - SE, alternative 5'/3' splice site –A5SS/A3SS, intron retention – IR, mutually exclusive exons – MXE). (B) Venn graph in the top right showing the overlap of differentially alternatively spliced AS

*Figure 3 continued on next page*

*Figure 3 continued*

events among the three circadian neuron subtypes versus TH neurons, as well as the number of genes involved in these AS events. Most of the transcripts undergoing differential AS events are unique to one subset of the circadian neurons. Only eight transcripts in eight genes show differential AS events in all three circadian neuron groups. Venn graph in the lower left showing the overlap of differentially expressed RBPs (potential splicing regulators) in each of the three circadian neuron subtypes compared to TH neurons. Very limited overlap is also found for the differentially expressed RBPs among neuron subpopulations. The RBPs Syp and mub which are differentially expressed in LNvs versus TH neurons target 32 and 23 LNv-specific AS event transcripts as determined from publicly available CLIP and RIP-seq experiment datasets (*Stoiber et al., 2015*).

DOI: https://doi.org/10.7554/eLife.35618.008

The following source data and figure supplements are available for figure 3:

**Source data 1.** Differentially spliced AS events in DN1 neurons versus the non-circadian TH neurons.
DOI: https://doi.org/10.7554/eLife.35618.012
**Source data 2.** Differentially spliced AS events in LNd neurons versus the non-circadian TH neurons.
DOI: https://doi.org/10.7554/eLife.35618.013
**Source data 3.** Differentially spliced AS events in LNv neurons versus the non-circadian TH neurons.
DOI: https://doi.org/10.7554/eLife.35618.014
**Source data 4.** Differentially spliced AS events in DN1 neurons versus LNd neurons.
DOI: https://doi.org/10.7554/eLife.35618.015
**Source data 5.** Differentially spliced AS events in LNd neurons versus LNv neurons.
DOI: https://doi.org/10.7554/eLife.35618.016
**Source data 6.** Differentially spliced AS events in DN1 neurons versus LNv neurons.
DOI: https://doi.org/10.7554/eLife.35618.017
**Source data 7.** Differentially expressed gene transcripts in DN1 neurons versus the non-circadian TH neurons.
DOI: https://doi.org/10.7554/eLife.35618.018
**Source data 8.** Differentially expressed gene transcripts in LNd neurons versus the non-circadian TH neurons.
DOI: https://doi.org/10.7554/eLife.35618.019
**Source data 9.** Differentially expressed gene transcripts in LNv neurons versus the non-circadian TH neurons.
DOI: https://doi.org/10.7554/eLife.35618.020
**Figure supplement 1.** Overlap of differentially spliced AS events in circadian neurons compared to non-circadian, TH neurons in each conventionally recognized splicing pattern category.
DOI: https://doi.org/10.7554/eLife.35618.009
**Figure supplement 2.** Identification of differentially spliced AS events among the circadian neuron subpopulations.
DOI: https://doi.org/10.7554/eLife.35618.010
**Figure supplement 3.** Functional comparison of differentially spliced cassette exons and non-differentially spliced cassette exons in circadian versus non-circadian TH neurons.
DOI: https://doi.org/10.7554/eLife.35618.011

are more prone to preserve reading frame than non-alternatively spliced exons (*Figure 3—figure supplement 3B*), further indicating that these AS events are important to the identity and function of the neuron subpopulation.

## Identification of cycling alternatively spliced isoforms in circadian neurons

To further investigate the role of AS regulation in circadian rhythms, we used JUM to identify AS events that exhibit cycling alternative splicing in each of the neuronal subtypes. To do this, we queried AS structures identified in each of the neuronal subgroups across the six time-point RNA-seq data (two independent replicas for each time point) to profile the changes in sub-AS- junction usage throughout the day (F24 and JTK-cycle; see Materials and methods). We identified cycling AS structures in all 3 groups of circadian neurons: 173 AS structures in DN1 neurons (5.7% of all AS structures), 92 in LNv neurons (5.0% of all AS structures), and 48 in LNd neurons (3.7% of all AS structures) (*Figure 6—source data 1*). These events affect 147, 81, 43, and nine genes in DN1, LNv, LNd and TH neurons, respectively (*Figure 6A*). Importantly, 85% of the transcripts that exhibit time-dependent changes in alternative splicing do not cycle at the total mRNA level (*Figure 6A*; RPKM based on ESAT quantification [*Derr et al., 2016*; *Abruzzi et al., 2017*]). The non-circadian TH neurons serve as a negative control for this analysis (only 11 cycling AS structures) because they have very few cycling mRNAs and probably do not express the clock genes (*Abruzzi et al., 2017*).

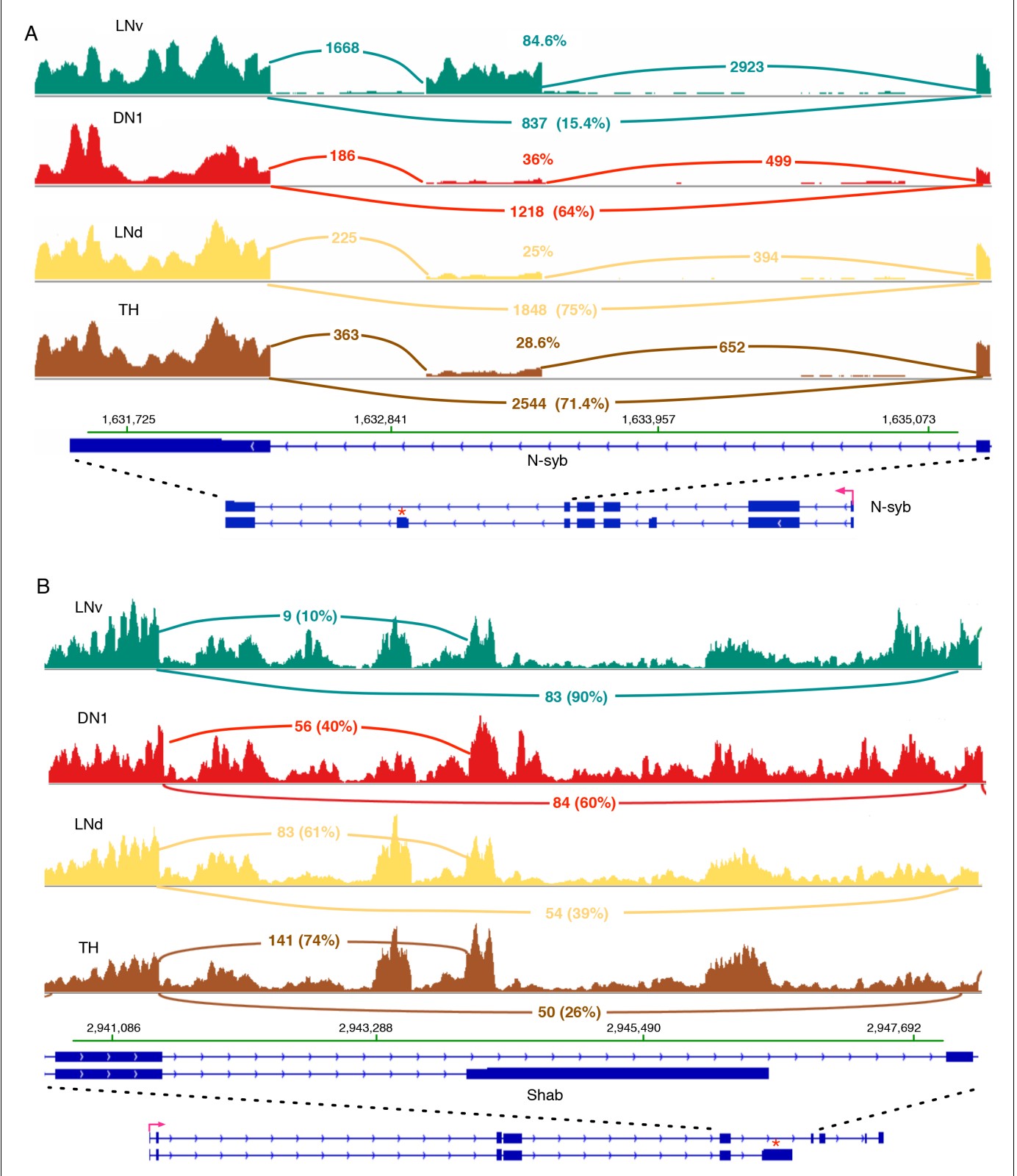

**Figure 4.** *N-syb* and *Shab* undergo differential splicing in circadian neurons compared to non-circadian, TH neurons. (**A**) In LNvs, the Neuronal synaptobrevin (N-syb) transcript is enriched for the isoform that includes an exon present in variant J. The alternatively spliced exon is marked by '*'. Use of this exon results in a N-syb protein with a different C-terminus (85 amino acids). (**B**) The transcript of the potassium-gated voltage channel, Shab, undergoes differential alternative splicing in LNvs and DN1s. This results in the higher usage of the terminal exon present in variant K of Shab (marked

*Figure 4 continued on next page*

*Figure 4 continued*

by '*'). This creates a protein with a novel C-terminus that lacks the sixth transmembrane region of the potassium channel. The high intronic signal seen in this region of Shab is due to the presence of a gene in this region on the opposite strand.

DOI: https://doi.org/10.7554/eLife.35618.021

The majority of these cycling AS structures are specific to a particular group of circadian neurons (*Figure 6B*). Some of this specificity is because certain AS structures are only found in one subgroup of circadian neurons (~40% in DN1, 14% in LNd and 25% in LNv). Moreover, only two transcripts show cycling alternative splicing in all three groups of circadian neurons: *still life* (*sif*) that regulates synaptic differentiation and the type II Camp-dependent protein kinase, *pKa-R1* (*Figure 6B*). This striking specificity suggests that time-of-day changes in alternative pre-mRNA splicing acts as an additional and previously unappreciated level of circadian regulation in *Drosophila* clock neurons, as previously suggested in mammals based on microarrays (*McGlincy et al., 2012*).

One striking example of a cycling AS structure is a transcript isoform encoding circadian kinase, Shaggy (from the fly gene *sgg*; known in mammals as GSK3). Alternative splicing of the *sgg* transcript is due to the use of two different 3'- splice sites generating distinct transcript and protein isoforms differing at the C terminus: a shorter isoform utilizing terminal Exon b and a longer isoform utilizing terminal Exon a that includes a specific additional 63 amino acids (*Figure 6C*; marked by '*' and '**', respectively). In LNv and DN1 neurons, the longer *sgg* isoform is almost exclusively used (*Figure 6—figure supplement 1*). However, in LNd neurons, the longer isoform is utilized almost exclusively in the morning (ZT0-ZT11) and then by late night (ZT18-24) there is approximately equal usage of the long and short *sgg* transcripts (*Figure 6C*; two replicates quantified in *Figure 6D*, red line and circle).

The potassium channel SLOWPOKE is encoded by the *Slo* gene and also undergoes neuron-specific AS cycling. There are two mutually exclusively used exons in the second alternatively spliced exonic regions in the *Slo* transcript: exon 2a and exon 2b that alter the cytosolic face of the channel (*Figure 6—figure supplement 2A*, marked by '*' and '**', respectively). In DN1 neurons only, exon 2b is included in the mornings and then there is a shift toward exon 2a inclusion in the late nighttime (*Figure 6—figure supplement 2A,B*). SLOWPOKE has been shown to affect circadian behavior in *Drosophila* (*Ceriani et al., 2002*), and its mammalian homolog, Kcnma1, shows similar phenotypes in mammals (*Meredith et al., 2006*). It is possible that the different protein isoforms created by time-of-day changes in alternative splicing mRNA isoforms are important for these functions (see Discussion).

To explore how cycling AS could impact neuronal functions, we performed gene ontology analysis on transcripts undergo cycling AS in the circadian neurons. Interestingly, modulation of locomotor behavior is enriched for cycling AS transcripts in all three circadian neuron groups, although a distinct group of transcripts is implicated in each circadian neuronal group (*Figure 6—figure supplement 3*; *Supplementary file 3*). In LNvs and LNds neurons, the most significantly enriched GO term for cycling AS transcripts is male courtship behavior and olfactory learning, respectively (*Figure 6—figure supplement 3A,B*; *Supplementary file 3*). In DN1 neurons, the top enriched GO term for cycling AS transcripts is mRNA binding proteins and 8 out of the 12 genes in this category encode alternative splicing regulators. The cycling AS of these splicing regulators could further facilitate downstream time of day-dependent changes in global splicing profiles (*Figure 6—figure supplement 3C*; *Supplementary file 3*).

Previous analysis of mRNA expression cycling in the circadian neurons revealed that mRNAs cycle with distinct phases in each subgroup of neurons (*Abruzzi et al., 2017*). Interestingly, the phases of the cycling AS structures also show neuron-group specific distributions (*Figure 7A*; *Figure 7—figure supplement 1*; blue). Most interesting were the DN1 neurons in which the majority of the time-of-day-dependent cycling AS structures peak in the late night (*Figure 7A*; blue). This peak is almost anti-phase to that of DN1 mRNA cycling (*Figure 7A*, red). In contrast, mRNA cycling and AS cycling show similar phases in the LNv neurons (*Figure 7—figure supplement 1A*). The LNd neurons have a third pattern: cycling mRNAs peak in the early evening, but cycling AS structures have a random distribution throughout the day (*Figure 7—figure supplement 1B*).

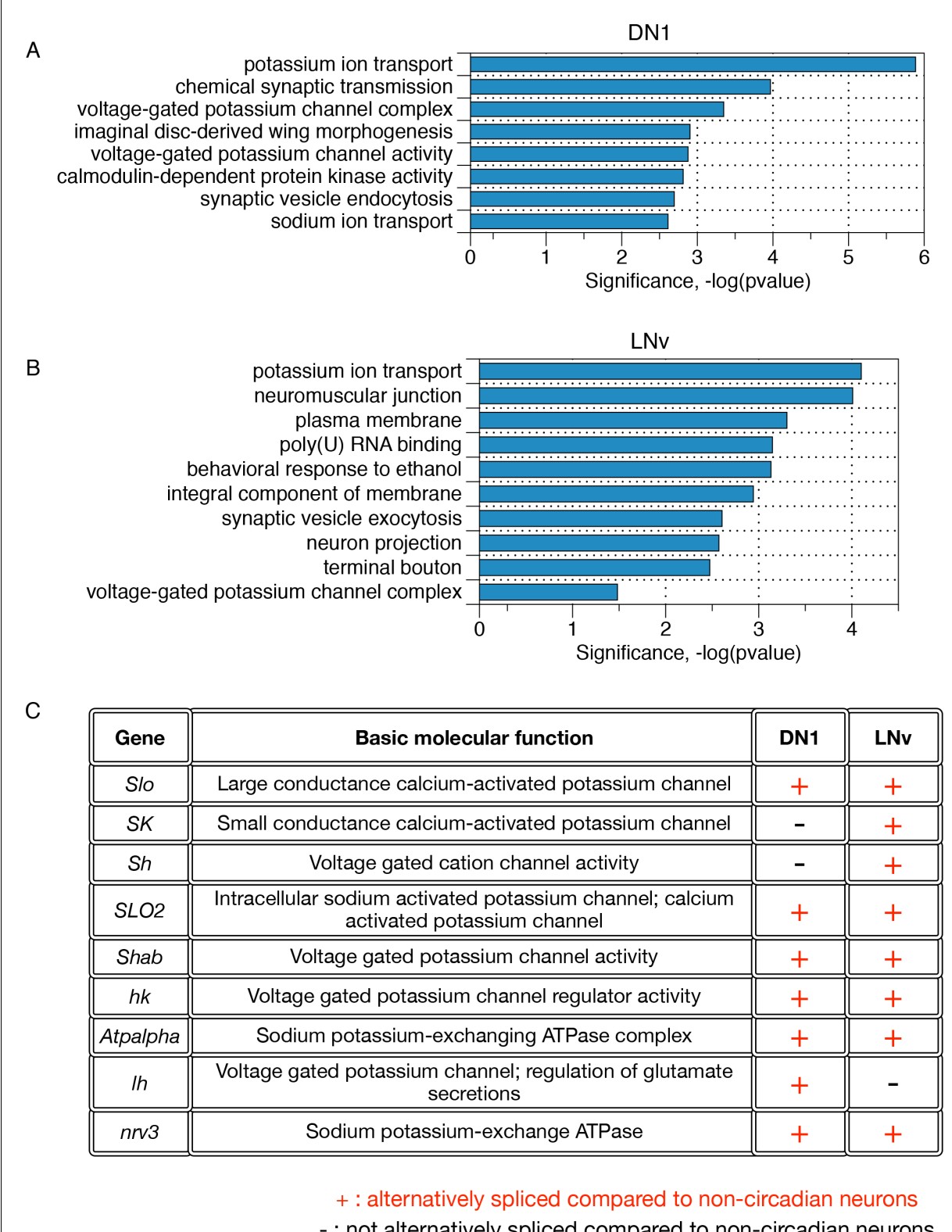

**Figure 5.** Differentially alternatively spliced transcripts in LNvs and DN1s compared to non-circadian neurons reflect specific neuronal functions. (**A**) Gene ontology analysis of those transcripts that undergo differential alternative splicing in DN1s reveals enrichments in potassium ion transport, calmodulin-dependent protein kinases, and synaptic vesicle endocytosis. (**B**) Gene Ontology (GO) analysis of those transcripts that undergo differential alternative splicing in LNvs reveals enrichments in potassium ion transport, synaptic vesicle exocytosis, neuron projections and terminal boutons. (**C**)

*Figure 5 continued*

Many differentially spliced transcripts in circadian neurons encode proteins involved in potassium ion transport, including six different classes of potassium channels as well as two sodium-potassium-exchanging ATPases.

DOI: https://doi.org/10.7554/eLife.35618.022

To probe potential mechanisms contributing to this neuron-specific AS cycling, we queried the list of cycling mRNAs within these neurons (*Abruzzi et al., 2017*) and identified cycling transcripts encoding RBPs in all three neuronal subgroups (*Figure 7B*). None of the identified RBP mRNAs cycle in more than one circadian neuron subgroup. Although there was no clear global correlation between the phases of these RBP mRNAs and cycling AS in any neuronal group, we did identify a cycling splicing factor in LNds, Qkr54B, that has been shown to target Sgg transcripts (Stoiber et al., 2015). Interestingly, Qkr54B mRNA peaks 3–4 hr prior to the time of day dependent inclusion of the alternative spliced *Sgg* terminal Exon b (*Figure 6D*; green line and triangle), suggesting that cycling RBP expression might contribute to the cycling AS patterns.

## Discussion

Tissues of the nervous and germline systems, such as brain, testes and ovaries, have more complex transcriptomes than other cell types due to extensive alternative pre-mRNA splicing or AS (*Wang et al., 2008*; *Pan et al., 2008*). The nervous system especially exhibits vast numbers of AS isoforms, many of which are novel and are only beginning to be comprehensively identified (*Li et al., 2007*; *Irimia et al., 2014*). This increase in transcript isoform complexity likely contributes to the specification and functional diversity of cell types within the nervous system.

Here, we have applied a novel computational algorithm called JUM to characterize the transcript isoform diversity generated by alternative splicing in three circadian neuronal subtypes (LNv, LNd and DN1), as well as a non-circadian dopaminergic neuron population (TH neurons) of the *Drosophila* central nervous system. JUM can comprehensively analyze, quantitate and compare tissue- or cell-type-specific AS patterns without requiring a priori annotations of known transcripts or transcriptomes (*Wang and Rio, 2017a*). Our analysis revealed a previously unappreciated diversity and complexity of alternatively spliced transcript isoform patterns in these four neuronal subtypes, suggesting that they contribute to neuronal identity, connectivity, activity and circadian functions. This is because many of these novel, previously undetected and unannotated isoforms were unique to a given neuronal population and occurred in transcripts from genes implicated in neuronal activity or circadian rhythms.

For example, the kinase Shaggy and the blue light photoreceptor cryptochrome play central roles in circadian clock regulation and have novel AS patterns in discrete subsets of the circadian neurons. In addition, nine different transcripts involved in potassium transport undergo differential AS in circadian neurons compared to non-circadian neurons. These transcripts encode six different potassium channels (*Figure 5*). Many of these genes have a complex organization known to encode populations of functionally distinct proteins isoforms, which change the activation kinetics as well as calcium sensitivity of the channels (*Johnson et al., 2011*). Neuronal firing is known to play a key role in the circadian circuit with recent studies illustrating that different subgroups of circadian neurons have characteristic time-of-day neuronal firing patterns (*Flourakis et al., 2015*; *Liang et al., 2016*; *Guo et al., 2017*). Although it is not yet fully understood which potassium channels play a critical role in each circadian neuron subgroup, several channel pre-mRNAs that undergo differential splicing in circadian neurons impact circadian behavior and sleep, such as slowpoke (Slo [*Jaramillo et al., 2004*]), Shaker (Sh; [*Cirelli et al., 2005*; *Pimentel et al., 2016*]) and Hyperkinetic (Hk; [*Fogle et al., 2015*]). It is therefore likely that AS adds diversity and distinct physiological properties to these protein isoforms, which then impacts neuron-specific firing patterns. From a more general perspective, AS augments transcriptional regulation in giving different circadian neurons individual identities and distinct functions.

Approximately 5% of the AS events identified in circadian neurons also undergo time-of-day dependent changes in alternative splicing (cycling splicing). It is important to note that all our experiments were conducted under 12 hr of light and 12 hr of dark conditions, making it impossible to distinguish between light and clock control. Nonetheless, these data indicate that splicing adds a

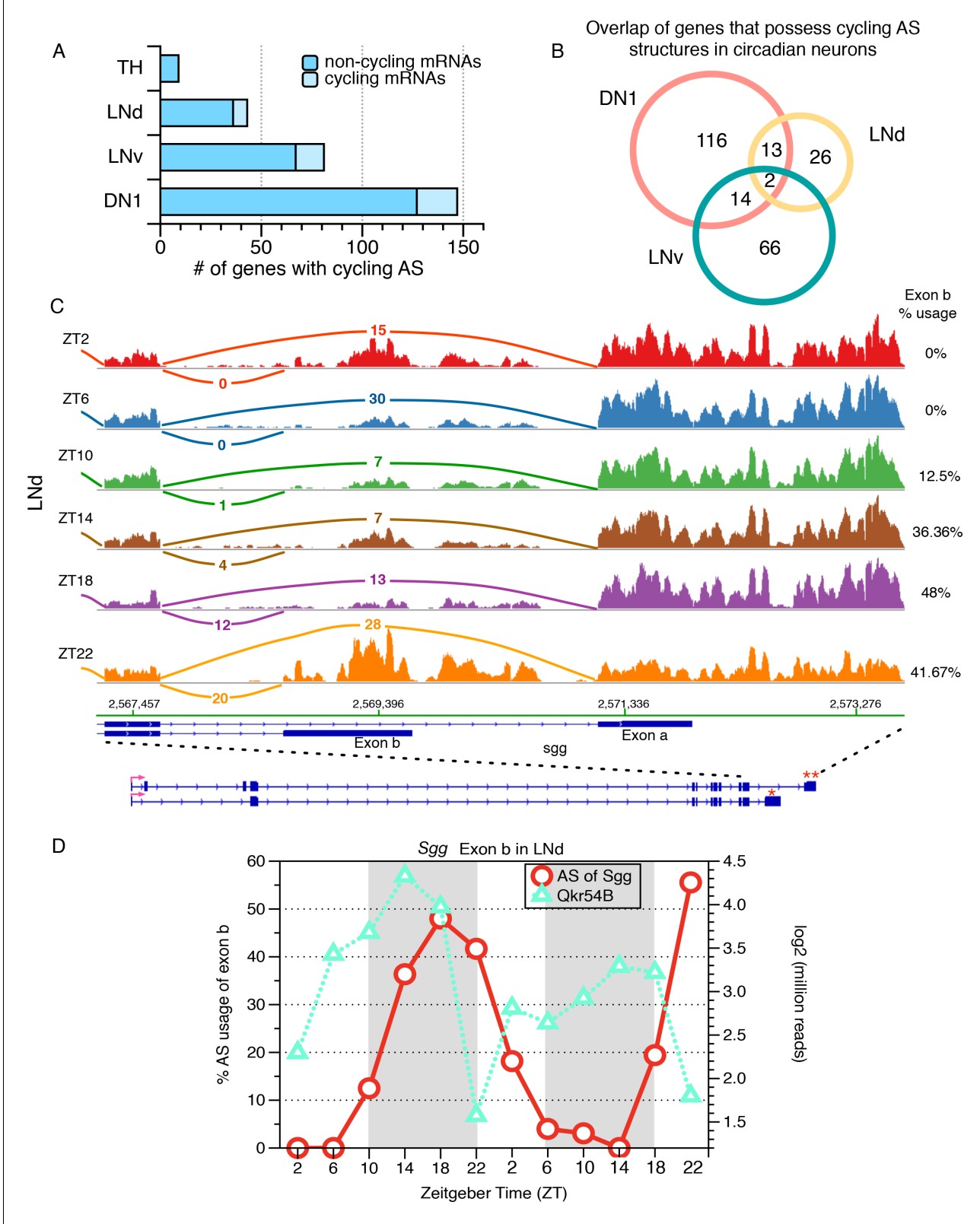

**Figure 6.** Identification of cycling alternative splicing in *Drosophila* circadian neurons. To identify cycling time-of-day changes in alternative splicing, the frequency at which each sub-AS-junction is utilized in every profiled AS structure was examined across six timepoints taken at four hour intervals throughout the day. Two independent six-timepoint-datasets were examined. (**A**) The distribution of the number of genes with cycling AS structures identified in each neuronal group. The majority of the cycling AS structures identified occur in non-cycling mRNAs (dark blue). As a non-circadian

*Figure 6 continued on next page*

*Figure 6 continued*

neuronal group, the number of genes with cycling AS identified in TH neurons serves as a negative control and estimate of false positives of the analysis. (B) Overlap of genes with cycling AS structures across the three circadian neuron subtypes. Most cycling AS structures are neuron specific; only two transcripts undergo cycling AS in all three groups of circadian neurons. (C) Sashimi plots highlight RNA-seq signal at the 3'-end of the gene encoding the circadian kinase, Shaggy (Sgg). The architecture of the Sgg gene is shown below; the red arrow indicates the direction of transcription. Time of day is indicated by Zeitgeber time (ZT) on the left side of the figure; ZT0 is the beginning of 12 hr of lights-on and ZT12 is the beginning of 12 hr of lights-off. The transcript of the circadian kinase, Shaggy (sgg), undergoes time-of-day oscillations in alternative splicing at the 3'-end of the transcript in LNd neurons. In early morning (ZT2/ZT6), terminal exon a (marked by '**') is preferred, generating a slightly longer protein isoform. In the night (ZT18/22), splicing shifts to utilize terminal exon b (marked by '*') that generates a shorter protein isoform with different C-terminus. (D) The frequency at which the shorter protein isoform of Sgg (with exon b) is generated is graphed throughout the day for two independent days, over two independent set of 6-timepoints in red circles, following the y axis on the left. The time-of-day dependent expression levels of the cycling Qkr54B transcript is shown using green triangles, following the y axis on the right. Qkr54B is a predicted regulator of Sgg transcripts determined by publically available RIP-seq experiments (Stoiber et al., 2015).

DOI: https://doi.org/10.7554/eLife.35618.023

The following source data and figure supplements are available for figure 6:

**Source data 1.** Cycling AS events in DN1, LNd, LNv and TH neurons.

DOI: https://doi.org/10.7554/eLife.35618.027

**Figure supplement 1.** Predominant inclusion of exon a of the *Sgg* transcript in LNv and DN1 neurons.

DOI: https://doi.org/10.7554/eLife.35618.024

**Figure supplement 2.** Identification of *slo* transcripts that exhibit cycling alternative splicing in *Drosophila* circadian neurons.

DOI: https://doi.org/10.7554/eLife.35618.025

**Figure supplement 3.** Transcripts with neuron subgroup-specific cycling AS structures reflect crucial neuronal functions.

DOI: https://doi.org/10.7554/eLife.35618.026

dramatic layer of gene regulation to diurnal changes in gene expression. Moreover, many of the cycling AS transcripts show constant overall mRNA levels, which suggests the existence of neuron-specific splicing factors that are expressed or activated only at specific times of the day. Indeed, we have identified several candidate cycling neuron-enriched transcripts that encode RBPs that may help to drive cycling AS patterns.

A recent trend in biological research is to generate transcriptome profiles from single cells. For example, this strategy is part of the 'human cell atlas' project aimed at personalized genomic medicine or the 'brain initiative' project to generate profiles of all neurons in the mouse brain (*Jorgenson et al., 2015*; *Regev et al., 2017*). One recent study was able to obtain about 20M sequence reads per isolated human iPS cell but only managed to analyze splicing patterns for the most highly expressed genes (*Song et al., 2017*). Our study in contrast used ~100 isolated *Drosophila* neurons for each of the four neuron subtypes along with judicious use of both oligo-dT and random hexamer priming of the cDNA libraries. This strategy obtained about 10–30M sequence reads for each sample, including substantial information from the 5' ends of transcripts, and JUM was able to detect and classify a large number of previously unannotated pre-mRNA isoforms. Many of them are missing from our fly head RNA-seq data assayed and analyzed in parallel, indicating that these new isoforms are cell-type specific. Not surprisingly, the novel isoforms from the three circadian neuron groups fall into many gene ontology (GO) categories associated with specific circadian clock activity and function.

Taken together, the work presented here indicates that the number of alternative splicing events that take place in neuronal tissues is grossly underestimated, even though publically-funded genome projects, such as the NIH modENCODE projects deeply sequenced transcriptomes from a variety of Drosophila tissues and developmental stages. This is despite the appreciation of how much AS occurs in the nervous system, for example recent comprehensive analysis of splicing patterns through deep sequencing of ~50 mouse and human tissues revealed about 2500 neuronally-regulated alternative splicing events (*Irimia et al., 2014*; *Li et al., 2015*). We therefore suggest that these events will need to be comprehensively evaluated by much deeper sequencing than is currently afforded by most contemporary single cell RNA-seq studies and by AS analysis software like JUM that is not constrained by a priori knowledge of known splicing events.

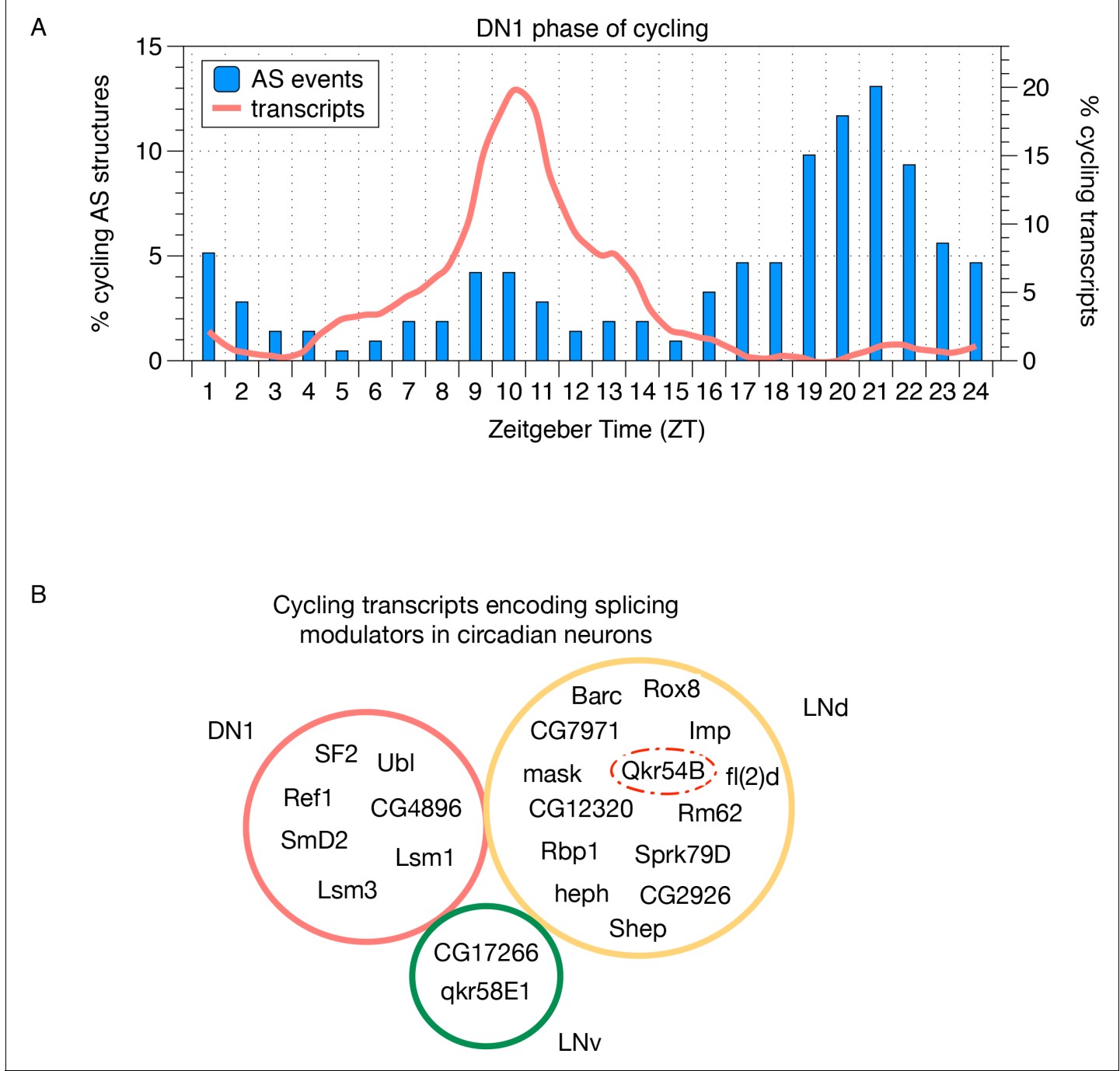

**Figure 7.** Cycling alternative splicing and mRNA expression in Drosophila circadian neurons. (**A**) The phase distribution of cycling AS structures in DN1 neurons is plotted as a histogram (blue) with the phase distribution of all cycling transcripts in DN1 neurons overlaid (orange). In DN1s, most cycling transcripts are at their highest level at mid-morning. In contrast, cycling AS structures are most abundant in the late night. (**B**) Cycling transcripts that encode RBPs that act as splicing modulators were identified. The Venn graph shows the overlap of these cycling RBPs in each of the circadian neuron subpopulations. Every circadian neuron expresses a unique set of cycling RBPs. Qkr54B (red dashed circle) is known to target the *Sgg* transcripts (Stoiber et al., 2015) that present cycling AS patterns in LNd neurons.

DOI: https://doi.org/10.7554/eLife.35618.028

The following figure supplement is available for figure 7:

**Figure supplement 1.** Phase distribution of cycling AS structures in LNvs and LNds.

DOI: https://doi.org/10.7554/eLife.35618.029

# Materials and methods

**Key resources table**

| Reagent type or resource | Designation | Source or reference | Identifiers | Additional information |
|---|---|---|---|---|
| RNA-seq data | | (*Abruzzi et al., 2017*) | GSE77451 | |
| RNA-seq data | | (*Wang et al., 2016*) | GSE79916 | |
| Software, algorithm | Morpheus | N/A | | https://software.broadinstitute.org/morpheus/ |
| Software, algorithm | BoxPlotR | (*Spitzer et al., 2014*) | | http://shiny.chemgrid.org/boxplotr/ |
| Software, algorithm | STAR 2.5.3a | (*Dobin et al., 2013*) | | https://github.com/alexdobin/STAR |
| Software, algorithm | Samtools 1.4.1 | (*Li, 2011*) | | http://samtools.sourceforge.net/ |
| Software, algorithm | Bedtools 2.26.0 | N/A | | http://bedtools.readthedocs.io/en/latest/ |
| Software, algorithm | JUM 1.3.7 | (*Wang and Rio, 2017b*) | | https://github.com/qqwang-berkeley/JUM (copy archived at https://github.com/elifesciences-publications/JUM) |
| Software, algorithm | ImageJ 1.50i | (*Schneider et al., 2012*) | | https://imagej.nih.gov/ij/ |
| Software, algorithm | David Bioinformatics Resources 6.8 | (*Huang et al., 2009b*) | | https://david.ncifcrf.gov/home.jsp |
| Software, algorithm | IGV 2.3.91 | (*Robinson et al., 2011*; *Thorvaldsdóttir et al., 2013*) | | http://software.broadinstitute.org/software/igv/ |

## Neuron dissection, RNA extraction and RNA-seq library preparation

Neuron dissection, RNA extraction and library preparation were performed as described in (*Abruzzi et al., 2015*).

## RNA-seq data mapping for JUM

The sequencing data used in this study is publically-available at Gene Expression Omnibus (Accession numbers GSE77451 (neurons) and GSE79916 (heads)). RNA-seq reads were mapped to the *Drosophila* (dm3) genome using STAR (*Dobin et al., 2013*) in the 2-pass mode; the exact commands used are listed in *Supplementary file 5*. Only uniquely mapped reads are kept for downstream analysis. The mapping statistics for each sample are listed in *Supplementary file 4*. For pooled data analysis, fastq files from each time points were pooled before subjecting the pooled data to mapping as described above. The mapping parameters for splice junction profiling are set as default in STAR: the minimum overhang length for splice junctions on both sides are set to be the 30 bp for non-canonical motif junctions, and 12 bp for the canonical GT/AG (CT/AC), GC/AG (CT/GC), AT/AC (GT/AT) motif junctions, respectively. The minimum uniquely mapped read count per junction is set to be three for non-canonical motif junctions and one for the canonical motif junctions. The minimum allowed distance to other splice junctions' donor/acceptor is set to be 10 for non-canonical motif junctions and 0,5,10 for the canonical GT/AG (CT/AC), GC/AG (CT/GC), AT/AC (GT/AT) motif junctions, respectively. The maximum gap allowed for junctions are set to be as follows: junctions supported by one read can have gaps $\leq$ 50000 b; if supported by two reads then gaps $\leq$ 100000 b; if supported by three reads then gaps $\leq$ 200000; if supported by more than four reads then gaps $\leq$ alignIntronMax (see the STAR manual for details).

## Profiling of AS junctions

Splice junctions with alternative 5' or 3' splice sites in the sample were profiled from the pool of total STAR-identified splice junctions for each neuronal sample, and defined as AS junctions. The novelty of each AS junction was compared against the library of annotated junctions in the UCSC genome browser RefSeq transcriptome annotation (genome version: FB2017_05).

## Gene Ontology analysis

Gene Ontology analyses were performed using David Bioinformatics Resources 6.8 (https://david.ncifcrf.gov/home.jsp) (*Huang et al., 2009a*). For neuron-subgroup-specific analyses, a list of transcripts expressed at greater than five reads was used as a background data set. A p-value of less than 0.05 was required for a gene to be considered enriched in the dataset.

## Visualization

All RNA-seq data are visualized using IGV (*Thorvaldsdóttir et al., 2013*; *Robinson et al., 2011*) and the Sashimi plots tool (*Katz et al., 2015*). Visualized tracks are further organized using ImageJ (*Schneider et al., 2012*). All boxplots in this paper were plotted using BoxPlotR (*Spitzer et al., 2014*).

## Differential AS analysis between circadian and non-circadian neurons using JUM

For each comparison of a circadian neuron subtype (DN1, LNv and LNd) and the non-circadian, TH neuron subtype, total splice junctions that receive more than 10 unique mapped reads in both collapsed datasets in the circadian or non-circadian neuron subtype samples were pooled together, and AS structures were constructed based on the pooled splice junction set. The usage of each sub-AS-junction in every AS structure was calculated and AS structures with significantly differentially 'used' sub-AS-junctions between each of the circadian neuron subtypes and the non-circadian neuron subtype were profiled as specified (*Wang and Rio, 2017a*). All AS structures were then assembled into the five conventionally recognized AS patterns – cassette exon (SE), alternative 5'/3' splice sites (A5SS/A3SS), mutually exclusive exons (MXE), and intron retention (IR), as well as a Composite category, which represents inseparable combinations of the conventional AS patterns. Here we are only focusing on AS events in the conventional AS pattern categories. Only AS events that showed more than 10% of change in alternative splicing and had a differential splicing test statistical pvalue $\leq$ 0.05 were considered significantly differentially spliced AS events. For more details see (*Wang and Rio, 2017a*) and the GitHub page of JUM (*Wang and Rio, 2017b*): https://github.com/qqwang-berkeley/JUM (copy archived at https://github.com/elifesciences-publications/JUM).

The same procedure was performed to analyze differential AS patterns among circadian neuron subgroups as well.

## Differential gene expression analysis between circadian and non-circadian neurons

Differential gene expression analysis between each circadian neuron subgroup and the non-circadian TH neurons were performed using DESeq2 (*Love et al., 2014*). Adjusted-p value 0.05 was chosen as the statistical cutoff.

## Cross-species conservation analysis in cassette exons

Conservation scores (PhastCon) (*Siepel et al., 2005*) for each single base in the cassette exons for alignments of 26 insects with *D.melanogaster* was downloaded from the UCSC Genome browser and an average PhastCon score for each cassette exon was calculated and compared.

## Cycling AS analysis

For each neuron subtype, total splice junctions that receive more than 10 unique mapped reads in at least one time point from both replicas were pooled. AS structures were constructed based on the pooled splice junction set. The relative usage of each sub-AS-junction in every AS structure is calculated, and subject to analysis using fourier analysis (F24; [*Wijnen et al., 2005*]) and JTK-cycle (*Hughes et al., 2010*). To be considered cycling using fourier transformation the following cutoffs were used: F24 score greater than 0.5, >1.5 fold change in splice junction usage, and the average transcript reads greater than 10 for at least one timepoint in each independent dataset. JTK cycle identified transcripts as cycling that had >1.5 fold change in splice junction usage, the average transcript reads greater than 10 for at least one timepoint in each independent dataset, and a p-value cutoff of less than 0.05. Phase determination was done using fourier transformation.

## Acknowledgements

We thank the Rosbash lab and the Rio lab for helpful discussions. This work was supported by NIH R01GM097352 and NIH R35GM118121 (D Rio, PI), the Center for RNA Systems Biology at UC., Berkeley (NIH P50102706; J Cate, PI) and the Howard Hughes Medical Institute (M Rosbash, PI). QW is

supported by the Arnold O Beckman Postdoctoral Fellowship. The authors declare no competing financial interests.

## Additional information

### Funding

| Funder | Grant reference number | Author |
|---|---|---|
| Arnold and Mabel Beckman Foundation | Postdoctoral Fellowship | Qingqing Wang |
| Howard Hughes Medical Institute | | Michael Rosbash |
| National Institutes of Health | R01GM097352 | Donald C Rio |
| National Institutes of Health | R35GM118121 | Donald C Rio |
| National Institutes of Health | NIH P50102706 | Donald C Rio |

The funders had no role in study design, data collection and interpretation, or the decision to submit the work for publication.

### Author contributions

Qingqing Wang, Software, Formal analysis, Validation, Investigation, Visualization, Methodology, Writing—original draft, Writing—review and editing; Katharine C Abruzzi, Data curation, Formal analysis, Validation, Investigation, Writing—original draft, Writing—review and editing; Michael Rosbash, Conceptualization, Supervision, Funding acquisition, Writing—review and editing; Donald C Rio, Conceptualization, Supervision, Funding acquisition, Writing—original draft, Writing—review and editing

### Author ORCIDs

Qingqing Wang http://orcid.org/0000-0002-0836-1367
Katharine C Abruzzi https://orcid.org/0000-0003-3949-3095
Michael Rosbash http://orcid.org/0000-0003-3366-1780
Donald C Rio http://orcid.org/0000-0002-4775-3515

### Decision letter and Author response

Decision letter https://doi.org/10.7554/eLife.35618.041
Author response https://doi.org/10.7554/eLife.35618.042

## Additional files

### Supplementary files

• Supplementary file 1. The distribution of splice site type (including reverse strand) in the AS junctions detected in each neuron group/sample. The distribution of splice site type in the AS junctions (including reverse strand) detected in each neuron group/sample that are within a gene.
DOI: https://doi.org/10.7554/eLife.35618.030

• Supplementary file 2. Gene ontology analyses of those transcripts that present neuron subgroup specific novel ALT junctions that are not found in whole head samples for each neuron subpopulation.
DOI: https://doi.org/10.7554/eLife.35618.031

• Supplementary file 3. Gene ontology analyses of those transcripts that present neuron subgroup specific cycling AS structures.
DOI: https://doi.org/10.7554/eLife.35618.032

• Supplementary file 4. Mapping statistics of the neuronal RNA-seq data to the Drosophila genome (dm3).
DOI: https://doi.org/10.7554/eLife.35618.033

• Supplementary file 5. Commands for STAR mapping.
DOI: https://doi.org/10.7554/eLife.35618.034

• Transparent reporting form
DOI: https://doi.org/10.7554/eLife.35618.035

## Data availability

All data generated or analysed during this study are included in the manuscript and supporting files.

The following previously published datasets were used:

| Author(s) | Year | Dataset title | Dataset URL | Database, license, and accessibility information |
|---|---|---|---|---|
| Wang Q, Taliaferro JM, Klibaite U, Hilgers V, Shaevitz JW, Rio DC | 2016 | The PSI-U1 snRNP interaction regulates male mating behavior in Drosophila | https://www.ncbi.nlm.nih.gov/geo/query/acc.cgi?acc=GSE79916 | Publicly available at the NCBI Gene Expression Omnibus (accession no: GSE79916) |
| Abruzzi K, Chen X, Nagoshi E, Zadina A, Rosbash M | 2015 | RNA-seq analysis of Drosophila clock and non-clock neurons reveals neuron-specific cycling and novel candidate neuropeptides | https://www.ncbi.nlm.nih.gov/geo/query/acc.cgi?acc=GSE77451 | Publicly available at the NCBI Gene Expression Omnibus (accession no: GSE77451) |

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
