## [Decision Letter]

Thank you for submitting your article "Diversity and circadian cycling of alternative splicing in the transcriptomes of isolated *Drosophila* neuron populations" for consideration by *eLife*. Your article has been reviewed by three peer reviewers, and the evaluation has been overseen by a Reviewing Editor and a Senior Editor. The reviewers have opted to remain anonymous.

The reviewers have discussed the reviews with one another and the Reviewing Editor has drafted this decision to help you prepare a revised submission.

Summary:

In this manuscript from Wang et al., the authors re-analyzed previously published RNA-seq datasets for alternative splicing (AS) patterns comparing three neuronal populations in *Drosophila melanogaster* that are critical for circadian rhythms (DN1, LNd, and LNv) with a control group of TH neurons. They have also analyzed these same populations over the course of two days. Overall, this is a coherent paper, applying sophisticated analytical method to high throughput RNA-seq data to uncover the novel findings.

Using the comprehensive computational method JUM (Junction Usage model) to analyze splicing patterns revealed a striking number of AS junctions in the aforementioned neurons absent from the more standard *Drosophila* head RNA preparation. A significant portion (~15%) of these were exclusive to the individual neuron types and almost entirely were previously unannotated (~95%). Each neuron type was also shown to contain a broad range of unique AS junctions when compared to the TH group, and that these genes could impact neuronal function (ex. N-syb, Shab). GO analysis on the AS genes from the circadian neurons showed globally that these patterns are more prominent in potassium channels, ion exchange pumps (sodium-potassium ATPases), and, more importantly, functions such as vesicle exocytosis that are critical for circadian function. Finally, the authors conducted a 2-day time course dissection of these same neurons every 4 hours and analyzed AS splicing again using JUM. This experiment found unique AS junctions in all circadian neurons that cycled through the day/night; however, the overall level of mRNA from most of these genes did not change, demonstrating that the control of gene function is most likely due to AS rather than transcriptional control.

There are interesting insights from these experiments into the much greater role of alternative splicing in circadian rhythms, however, there is too much emphasis on JUM, which has already been described elsewhere, in Wang and Rio, 2017. Considering this, the extensive description of the tool, nomenclature, and usage is not necessary and detracts from the biological aspect of the manuscript. That is not to say that JUM is not worth a good amount of emphasis – it is impressive, but a revision is needed to achieve a better balance. Substantial portions of the Results section and relevant Figures (Figure 1A in particular) could be removed.

Essential revisions:

1) The authors indicate that they have generated RNA-seq data from individual neuronal populations as described in Abruzzi et al., 2017 but it is not clear if the flies were maintained in 12:12 light:dark or constant darkness during the experiment. This distinction could impact the interpretation of circadian analyses since under 12:12 light:dark many gene expression changes and AS events may be driven by light sensing as opposed to an intrinsic clock.

2) The authors consistently refer only to the number of AS junction sites, without providing the number of affected genes. While the number of events is of interest because it gives a sense of how pervasive alternative splicing is, it how many genes are affected is necessary to get a sense of the scope and selectivity of this phenomenon. For example, in the Results section the authors explain the AS junctions in two genes, Cry and CG10483, but do not provide a perspective – how many of the genes with these AS structures have functions relevant to the different neurons in this study and of neuronal populations over other cell types?

3) There were only 10 AS events found in all three groups of neurons (Figure 3B). Was any further analysis done on these? These might be genes that are particularly important as well.

4) The authors could point out the use of other algorithms to identify AS changes – why rely only on JUM?

5) A critical point of this study is the identification of "novel" AS junctions. Therefore, it will be important to have some quality control metrics to show the identified junctions are real rather than technical artifacts or alignment errors. It will be helpful to clearly describe parameters related to accuracy of exon junction identification (e.g., size of overlap on each side of the junction), and if possible provide an estimate of accuracy (e.g. using simulated synthetic exon junction reads by swapping 5' and 3' parts).

6) Related to the point above, it is also important to clearly state the criteria used to define a "novel" AS junction in the main text. As of now, the authors stated in the method section that "The novelty of each AS junction was compared against the library of annotated junctions in the UCSC genome browser transcriptome annotation (genome 422 version: FB2017_05)." It is unclear what type of annotations was used here (e.g., which gene models, does that include EST sequences?)

7) In subsection “Circadian neurons present specific alternative pre-mRNA splicing patterns”, the authors identified AS junctions differentially spliced in each type of circadian neurons compared to TH neurons. They then examined the overlap of the lists to claim most of these junctions are specifically spliced in one subtype. To be more rigorous, one should perform differential splicing analysis between different types of circadian neurons, as the limited overlap could also be due to limited statistical power.

Similar issue for comparison of cycling exons in Figure 6B.

8) While cell type- and time of day-dependent AS patterns are artfully presented here, a major weakness in this manuscript is that the underlying mechanisms driving AS or their consequences on protein expression are not studied. Regarding upstream mechanism, it would help to know if expression of any RNA binding proteins correlate with the observed AS events across cell types or times of day. Regarding the relative impact of AS on neuronal identity and function, a first pass analysis could be to compare the relative numbers and enriched ontologies between mRNA expression and AS differences between cell types and in the circadian data. A more compelling experiment would be to specifically inhibit an AS event within one of the neuronal populations to test its effect on cell function.

9) One potential concern is the technical/biological variation between replicates, which is expected from the small number of cells used to generate RNA-seq libraries and visible from the examples shown in the figures. It will be helpful to have some data to show the reliability/reproducibility of observed differential splicing (e.g., correlation of splicing pattern of replicates, or even independent RT-PCR validation if possible).

10) The observation of alternative exons that are differentially spliced in specific types of neurons or time-of-the-day during the circadian clock is very interesting. Is there any indication of corresponding changes in splicing factors that potentially regulate these AS events?

11) It will be helpful to provide additional functional analysis of identified AS events with differential splicing, in addition to GO. This reviewer understands that detailed functional dissection of specific events might be beyond the scope of this manuscript, but it should be feasible to perform statistical analysis of cross-species conservation, preservation of reading frame, exon size, etc., which is frequently distinct for exons with cell type specific splicing.

12) In Figure 4B, although the authors claim, "in TH and LNd neurons the Shab transcripts preferentially utilize an alternative terminal exon, […] while in LNv and DN1 neurons, the long Shab isoform that encodes all transmembrane regions is dominant", it is not intuitively showing that by looking at the Sashimi plots. The authors should order the samples similarly to Figure 4A, and also may better prove their point by grouping the LNv and DN1 samples or highlighting the relevant differences in the figure read numbers.

---

## [Author Response]

Summary:[…] There are interesting insights from these experiments into the much greater role of alternative splicing in circadian rhythms, however, there is too much emphasis on JUM, which has already been described elsewhere, in Wang and Rio, 2017. Considering this, the extensive description of the tool, nomenclature, and usage is not necessary and detracts from the biological aspect of the manuscript. That is not to say that JUM is not worth a good amount of emphasis – it is impressive, but a revision is needed to achieve a better balance. Substantial portions of the Results section and relevant Figures (Figure 1A in particular) could be removed.

The reviewers suggested that there is too much emphasis on the computational method (JUM) used to analyze the RNA-seq datasets in this work, since it has already been described in detail in our JUM method paper deposited in bioRxiv. We have modified the manuscript accordingly and only kept a concise description of the nomenclature of the method for easy reference of the readers.

Essential revisions:1) The authors indicate that they have generated RNA-seq data from individual neuronal populations as described in Abruzzi et al., 2017 but it is not clear if the flies were maintained in 12:12 light:dark or constant darkness during the experiment. This distinction could impact the interpretation of circadian analyses since under 12:12 light:dark many gene expression changes and AS events may be driven by light sensing as opposed to an intrinsic clock.

We (Rosbash lab) have been focusing on LD (light-dark) behavior since beginning these experiments with microarrays a decade ago with the first publications in 2010. More relevant of course are the LD data (Abruzzi et al., 2017) on which this analysis is based. So, we are of course constrained without generating this massive amount of data de novo in constant darkness. We took this strategy at the outset, which we still endorse, because fly behavior is richer in LD than in DD, which allows for a more interesting comparison between transcripts, their presence in specific neurons as well as their cycling, and behavior. Because the reviewer is certainly correct, and light is probably germane (light alone or an interaction between light and the clock), we have also avoided the term "circadian" for these transcripts but refer to them as "cycling" or "oscillating" given their once/day or diurnal pattern. Nonetheless, we now clarify this point explicitly in the Discussion section.

2) The authors consistently refer only to the number of AS junction sites, without providing the number of affected genes. While the number of events is of interest because it gives a sense of how pervasive alternative splicing is, it how many genes are affected is necessary to get a sense of the scope and selectivity of this phenomenon. For example, in the Results section the authors explain the AS junctions in two genes, Cry and CG10483, but do not provide a perspective – how many of the genes with these AS structures have functions relevant to the different neurons in this study and of neuronal populations over other cell types?

We have included additional data throughout the paper that address the number and identity of the transcripts undergoing alternative splicing. The number of genes affected has been added to Figure 1B,D and Figure 3A,B. Gene ontology analysis supporting Figure 1 and Figure 6 has been added as Figure 1—figure supplement 2 and Figure 6—figure supplement 3, as well as Supplementary file 2 and Supplementary file 3. Descriptions of these new data have been added to subsection “There are many novel AS patterns in neuron subpopulations”; subsection “Identification of cycling alternatively spliced isoforms in circadian neurons”.

3) There were only 10 AS events found in all three groups of neurons (Figure 3B). Was any further analysis done on these? These might be genes that are particularly important as well.

We did a detailed analysis of the significantly changed AS events comparing TH neurons to all three groups of circadian neurons. Interestingly, the specific AS changes occur in genes that are functionally enriched in regulating neuronal plasticity, remodeling and synaptic transmission (Pten, PRL-1, Pdp-1, Sap47, Rim, mmd, CG9945 and RpS3A). It is likely that AS serves as an independent layer of gene regulation to affect the diversity of synapses and synaptic plasticity in the circadian neurons. We added the results from this analysis per the reviewer’s request in subsection “Circadian neurons present specific alternative pre-mRNA splicing patterns”. In addition, we made minor revisions to the numbers shown in Figure 3B. The total overlapping differentially spliced AS events from all three groups of circadian neurons versus THs should be 8 instead of 10. Previously some header lines in the gene list files were counted towards the number of genes. These have been corrected, but the conclusions stay the same.

4) The authors could point out the use of other algorithms to identify AS changes – why rely only on JUM?

We chose the JUM algorithm to analyze the neuronal subpopulation datasets because of JUM’s unique feature for analyzing differential AS patterns completely independent of *a priori* knowledge of the host transcriptome annotation. It is well known that tissues such as neurons and gonads possess exceptionally diverse and dynamic cellular AS profiles associated with various cellular states and functions, greatly exceeding our current knowledge of the transcriptome annotations, especially in non-human model organisms such as *Drosophila*. Almost all other currently available methods for AS analysis rely on pre-built annotated AS splice junction libraries or transcriptome annotations, which result in vast underestimates of the tissue-specific AS complexity (Wang and Rio, 2017). We have specified the rationale for choosing JUM for AS analysis in subsection “There are many novel AS patterns in neuron subpopulations” in the revised manuscript. We have also explained the rationale and showed the superior performance of JUM on this particular point that the reviewer raised in detail in the preprint of the JUM paper (Wang and Rio, 2017).

5) A critical point of this study is the identification of "novel" AS junctions. Therefore, it will be important to have some quality control metrics to show the identified junctions are real rather than technical artifacts or alignment errors. It will be helpful to clearly describe parameters related to accuracy of exon junction identification (e.g., size of overlap on each side of the junction), and if possible provide an estimate of accuracy (e.g. using simulated synthetic exon junction reads by swapping 5' and 3' parts).

JUM accepts output files from the third-party RNA-seq alignment software tool STAR that performs read mapping and AS junction profiling with the following additional filter settings to control for the quality of AS junction detection and quantification: (1) JUM uses the two-pass alignment approach that has been shown to greatly improve novel splice junction quantification (Veeneman et al., 2016); (2) JUM only considers unambiguously mapped reads over splice junctions for junction profiling and quantification; and (3) JUM applies stringent quality filters for AS junctions. An AS junction will be taken for downstream analysis only if it receives more than 10 unambiguously mapped reads in every one of the biological replicates of the sample. These settings make sure that the AS junction analyzed by JUM has been observed consistently and reproducibly from the RNA-seq experiments. The accuracy of JUM-quantified novel AS junctions has also been tested by multiple rounds of qRT-PCR and RT-PCR validation experiments using experimental data from mouse embryonic cortical neurons and male fruit fly heads and the validation rates stay ~100% so far (Wang et al., 2013, Wang et al., 2016, Wang and Rio, 2017). In addition, JUM uses STAR as the aligner because STAR is among the best short-read alignment software developed so far for both comprehensiveness and accuracy (Baruzzo et al., 2017, Engstrom et al., 2013) and has been tested in numerous both simulated and experimental datasets on multi aspects of alignment performance. Given the extensive testing of STAR referenced above, estimating the performance of STAR using synthetic RNA-seq reads is beyond the scope of this paper.

We have indicated the mapping metrics used in this analysis for the data presented in this manuscript in subsection “RNA-seq data mapping for JUM” in the revised version as follows: “the minimum overhang length for splice junctions on both sides are set to be a default of 30 bp for non-canonical motif junctions, and 12 for the canonical GT/AG (CT/AC), GC/AG (CT/GC), AT/AC (GT/AT) motif junctions, respectively. The minimum uniquely mapped read count per junction is set to be the default 3 for non-canonical motif junctions and 1 for the canonical motif junctions. The minimum allowed distance to other junctions’ donor/acceptor is set to be the default 10 for non-canonical motif junctions and 0, 5, 10 for the canonical GT/AG (CT/AC), GC/AG (CT/GC), AT/AC (GT/AT) motif junctions, respectively. Maximum gap allowed for junctions are set to be follows: junctions supported by 1 read can have gaps <= 50000b; if supported by 2 reads then gaps <= 100000b; if supported by 3 reads then gaps <= 200000; if supported by more than 4 reads then gaps <= alignIntronMax.”

6) Related to the point above, it is also important to clearly state the criteria used to define a "novel" AS junction in the main text. As of now, the authors stated in the method section that "The novelty of each AS junction was compared against the library of annotated junctions in the UCSC genome browser transcriptome annotation (genome 422 version: FB2017_05)." It is unclear what type of annotations was used here (e.g., which gene models, does that include EST sequences?)

The AS junctions identified in this study were compared to the RefSeq gene transcriptome annotation database and it does not include ESTs. We have specified this in the manuscript, subsection “Profiling of AS junctions”.

7) In subsection “Circadian neurons present specific alternative pre-mRNA splicing patterns”, the authors identified AS junctions differentially spliced in each type of circadian neurons compared to TH neurons. They then examined the overlap of the lists to claim most of these junctions are specifically spliced in one subtype. To be more rigorous, one should perform differential splicing analysis between different types of circadian neurons, as the limited overlap could also be due to limited statistical power.Similar issue for comparison of cycling exons in Figure 6B.

We have performed differential AS analysis between the circadian neurons DN1, LNd and LNv. We found vastly different AS patterns among each circadian neuron subpopulation – a total of 454, 778, and 325 AS events are significantly differentially spliced between DN1 versus LNd neurons, DN1 versus LNv neurons and LNd versus LNv neurons, respectively. This result supports our findings in Figure 3 and in the main text that each circadian neuron group possesses its own distinct AS profiles. We have added the results from this analysis in subsection “Circadian neurons present specific alternative pre-mRNA splicing patterns” and in Figure 3—figure supplement 2. Furthermore, we have conducted more analyses to explore the potential regulatory mechanisms that contribute to the unique AS profiles in circadian neuron subpopulations (detailed more in our response to #8 below). The limited overlap in differentially spliced AS events compared to TH neurons in the circadian neuron groups could be due to the observation that each circadian neuron group expresses its own unique set of RNA-binding protein/splicing factors compared to TH neurons (results added in Figure 3B). For example, we found that two LNv-specific splicing regulators mub and Syp targets ~36% of the AS events that are specifically differentially spliced in LNv neurons versus TH neurons. We have now included all of these additional results in subsection “Circadian neurons present specific alternative pre-mRNA splicing patterns”. It also needs to be noted that the reason for our approach to first perform differential AS analysis between each circadian neuron subgroup and the non-circadian TH neuron and then examining the overlap of each comparison was because we intended to profile for AS events that specifically affect circadian versus non-circadian functions. A direct comparison among the circadian subpopulations does not provide information that directly leads to our original intention and will also reveal AS events that are specialized in other functions in the circadian neurons.

Regarding Figure 6B, it is not trivial to compare two sets of 12 timepoint data (2 days 6 timepoints each) and ask whether the cycling tracks from two circadian neuron groups are statistically different from one another, as cycling determination involves multiple parameters including the shape of the curve, period length, amplitude requirements as well as minimum read requirements. However, there are several possible explanations for the neuron-specific cycling alternative splicing. Some of the alternative splicing events that cycle are neuron-specific (~40% in DN1, 14% in LNd and 25% in LNvs), so they not only cycle but are also only found in one subgroup of neurons. A smaller portion of alternative splicing events are present in all neuron groups yet cycle only in 1 (~20% in all neuron groups). In addition, we also found that each circadian neuron subpopulation possesses its own unique profile of cycling RNA-binding proteins/splicing factors that can potentially contribute to the neuron-specific cycling alternative splicing observed (Figure 7B). We have now included these additional results in subsection “Identification of cycling alternatively spliced isoforms in circadian neurons”.

8) While cell type- and time of day-dependent AS patterns are artfully presented here, a major weakness in this manuscript is that the underlying mechanisms driving AS or their consequences on protein expression are not studied. Regarding upstream mechanism, it would help to know if expression of any RNA binding proteins correlate with the observed AS events across cell types or times of day. Regarding the relative impact of AS on neuronal identity and function, a first pass analysis could be to compare the relative numbers and enriched ontologies between mRNA expression and AS differences between cell types and in the circadian data. A more compelling experiment would be to specifically inhibit an AS event within one of the neuronal populations to test its effect on cell function.

We agree that understanding the mechanisms behind neuron group-specific splicing is important and interesting. Inhibition of a particular AS event in one of the neuronal populations would be technically challenging. However, we have addressed this question as the reviewer suggests using a more extensive analysis of the RNA-seq data from the isolated neuronal populations, as well as publicly available databases of the targets of RNA-binding proteins (through CLIP and RIP experiments).

First, we profiled RNA binding proteins (RBPs) that are differentially expressed in each of the circadian neurons compared to the non-circadian TH neurons, respectively (New Figure 3B, and subsection “Circadian neurons present specific alternative pre-mRNA splicing patterns”). Interestingly, each circadian neuronal subpopulation expresses its own unique set of differentially-expressed RBPs compared to TH neurons, with very limited overlap, which is correlated with the observed unique differentially spliced AS patterns in each of the circadian neuronal populations compared to the AS patterns in TH neurons. We further profiled the targets of several of these RBPs using publicly-available data (Stoiber et al., 2015, Hansen et al., 2015) and found that these neuron-specific RBPs potentially target many AS events observed in each circadian neuronal subpopulation versus TH neurons (subsection “Circadian neurons present specific alternative pre-mRNA splicing patterns”). For example, Syb and mub, two RBPs that are specifically differentially-expressed in LNv neurons compared to TH neurons, target 23 and 32 LNv-specific differentially spliced AS events compared to TH neurons, respectively, with 5 overlapped AS events (~36% of total LNv neuron specific differentially spliced AS events). Although not definitive, this analysis supports the possibility that circadian neuron-specific RBP expression could drive the observed circadian neuron-specific AS patterns compared to TH neurons.

Second, we have performed a similar analysis to identify RBPs involved in splicing that cycles in the circadian neuron subgroups (New Figure 7B and Discussion section). Interestingly, each of the circadian neuron groups have distinct sets of cycling splicing-related transcripts that could contribute to the highly specific cycling splicing patterns observed in each circadian neuron group (Figure 7B). For example, the RNA-binding protein Qkr54B cycles in LNd neurons and one of its reported targets is Sgg (Stoiber et al., 2015), which displays cycling AS also in LNd neurons (Figure 6D). Furthermore, we examined the phase of all cycling AS structures in the three circadian neuron groups. Interestingly, each circadian neuron group exhibits distinct cycling phase patterns (Discussion section). Cycling AS structures in LNd cells peak throughout the day, while both DN1 and LNv cells showed higher levels of cycling AS structures at certain times of the day. DN1 cycling AS structures peak maximal in the late night and LNv cycling AS structure peaks show a bimodal distribution in mid-day and mid-night (Figure 7A and Figure 7—figure supplement 1). Moreover, some (but not all) of the cycling splicing factors found in each neuronal group showed a coordinated phase with their targeted cycling AS events. For example, the cycling RBP Qkr54B mRNA peaks 3-4 hours prior to the time of day dependent cycling inclusion of exon b in its targeted AS transcript Sgg (Figure 6D).

Third, we have compared the enriched gene ontologies between mRNAs and AS events that are differentially expressed or spliced, respectively, in each of the circadian neuron groups versus TH neurons. These comparisons suggest that neuron-specific differences in gene expression and splicing are impacting neuronal identity differently. For example, only one gene ontology category, integral membrane components, is found in common between the differentially spliced AS events and differentially expressed mRNA events in LNv neurons. However, neuropeptide receptors are distinctively enriched in differentially expressed mRNAs while potassium ion channels are distinctively enriched in differentially spliced AS splicing events in LNv neurons.

Lastly, we have used a similar approach to compare GO terms of neuron-specific cycling mRNAs and cycling AS structures. There is very little overlap between the types of genes that cycle in each of the circadian neuron groups at the mRNA level and at the level of splicing. The one exception is that the GO term chemical synaptic transmission is found in both cycling mRNAs and cycling AS structures in LNd neurons. One interesting observation is that the top enriched GO term for mRNAs that undergo cycling alternative splicing in DN1 neurons are mRNA binding proteins themselves (Figure 6—figure supplement 3 and Supplementary file 3). This list includes 8 known factors that impact splicing and the time-of-day dependent alternative splicing of these transcripts could influence AS profiles in DN1s.

These results above suggest that the neuronal-specific RBP expression and cycling patterns can be an important factor that drive the circadian neuron-specificAS profiles and cycling observed. That said, many other factors can contribute to the AS changes and patterns observed in each of the neuronal subpopulations. Possible mechanisms include, but not limited to, differential signaling pathway inductions, post-translational modifications and activation of splicing regulators and chromatin modifications. We believe these can be interesting and crucial topics for future studies that beyond the scope of this manuscript. The results above also suggest that AS and transcriptional regulation are two independent layers of gene regulation in specifying neuronal identify and function.

9) One potential concern is the technical/biological variation between replicates, which is expected from the small number of cells used to generate RNA-seq libraries and visible from the examples shown in the figures. It will be helpful to have some data to show the reliability/reproducibility of observed differential splicing (e.g., correlation of splicing pattern of replicates, or even independent RT-PCR validation if possible).

For software tools like MISO and MATS that can only perform pair-wise comparisons and cannot handle biological replicates, a correlation analysis of splicing patterns from replicates is indeed necessary. However, JUM integrates biological variability from multiple replicates into its core statistical model to make sure that the reported significantly changed AS events are supported by all replicates consistently and reproducibly. JUM achieves this by modeling the reads that map to an AS junction as a negative binomial distribution and fitting a mean-variance function through an inferred dispersion parameter with an empirical Bayesian approach across the observed variability from all replicates. This approach has been described previously, optimized recently and been widely applied in software tools analyzing RNA-seq experiments with replicates, and has been proven to greatly improve the quality of analysis by taking biological replicates into account (Anders et al., 2012, Love et al., 2014, Lu et al., 2005, McCarthy et al., 2012, Robinson et al., 2010, Robinson and Smyth, 2007, Robinson and Smyth, 2008). As a result, the nature of the model design in JUM already takes into account the reliability/reproducibility issues raised here by the reviewer. We also want to point out that we have consistently validated all JUM-reported AS events using multiple rounds of qRT-PCR in mouse embryonic cortical neurons and male fruit fly heads (Wang et al., 2013, Wang et al., 2016, Wang and Rio, 2017). This experimental validation further supports the accuracy of JUM’s predictions.

10) The observation of alternative exons that are differentially spliced in specific types of neurons or time-of-the-day during the circadian clock is very interesting. Is there any indication of corresponding changes in splicing factors that potentially regulate these AS events?

We have addressed this question in detail in our response to #8, above.

11) It will be helpful to provide additional functional analysis of identified AS events with differential splicing, in addition to GO. This reviewer understands that detailed functional dissection of specific events might be beyond the scope of this manuscript, but it should be feasible to perform statistical analysis of cross-species conservation, preservation of reading frame, exon size, etc., which is frequently distinct for exons with cell type specific splicing.

We have performed statistical analyses to compare cross-species conservation and preservation of reading frames in the cassette exons that are alternatively spliced versus the ones that are not alternatively spliced between DN1, LNd and LNv neurons versus the non-circadian TH neurons, respectively. For each comparison of a circadian neuron subpopulation versus TH, we compared the PhastCons conservation score of each identified cassette exon across 27 species of insects and also the size of the cassette exon to see if the alternative inclusion of the cassette exon changes the reading frame of the transcript. We found that neuron-subgroup-specific, alternatively spliced cassette exons are somewhat more conserved (although not statistically significant) and tend to preserve the reading frame of the transcript, indicating these alternatively spliced cassette exons are under more evolutionary selection to preserve their functions, which may be crucial to the identity and function of the neuronal subpopulations. These results are listed in Figure 3—figure supplement 3 and subsection “Circadian neurons present specific alternative pre-mRNA splicing patterns” in the revised manuscript.

12) In Figure 4B, although the authors claim, "in TH and LNd neurons the Shab transcripts preferentially utilize an alternative terminal exon, […] while in LNv and DN1 neurons, the long Shab isoform that encodes all transmembrane regions is dominant", it is not intuitively showing that by looking at the Sashimi plots. The authors should order the samples similarly to Figure 4A, and also may better prove their point by grouping the LNv and DN1 samples or highlighting the relevant differences in the figure read numbers.

We have highlighted the relevant differences in the figure read numbers in Figure 4B as the reviewer suggested.